# HyResPINNs: A Hybrid Residual Physics-Informed Neural Network Architecture Designed to Balance Expressiveness and Trainability

**Madison Cooley**  *mcooley@sci.utah.edu*
*Scientific Computing Institute*
*University of Utah*
*Salt Lake City, UT*

**Mike Kirby**  *kirby@sci.utah.edu*
*Scientific Computing Institute*
*University of Utah*
*Salt Lake City, UT*

**Shandian Zhe**  *zhe@utah.edu*
*Kahlert School of Computing*
*University of Utah*
*Salt Lake City, UT*

**Varun Shankar**  *shankar@cs.utah.edu*
*Kahlert School of Computing*
*University of Utah*
*Salt Lake City, UT*

**Reviewed on OpenReview:** *https://openreview.net/forum?id=et9WkjkqAw*

## Abstract

Physics-informed neural networks (PINNs) have emerged as a powerful approach for solving partial differential equations (PDEs) by training neural networks with loss functions that incorporate physical constraints. In this work, we introduce HyResPINNs, a two-level convex-gated architecture designed to maximize approximation expressiveness for a fixed number of degrees of freedom (DoF). The first level involves a trainable, per-block combination of smooth basis functions with trainable sparsity, and deep neural networks; the second involves the ability to gate entire blocks (much like in ResNets or Highway Nets), allowing for expressivity along the depth dimension of the architecture. Our empirical evaluation on a diverse set of challenging PDE problems demonstrates that HyResPINNs consistently achieve superior accuracy to baseline methods while remaining competitive relative to training times. These results highlight the potential of HyResPINNs to combine desirable features from traditional scientific computing methods and modern machine learning, paving the way for more robust and expressive approaches to physics-informed modeling.

## 1 Introduction

Partial differential equations (PDEs) underpin the simulation of complex physical phenomena across physics and engineering, from turbulent flows to electromagnetic waves and quantum systems (Evans, 2010; LeVeque, 2007). However, the interplay of nonlinearities, intricate geometries, and temporal dynamics often renders analytic solutions intractable, motivating a need for robust and adaptable numerical methods (Strang et al., 1974; LeVeque, 2007). While classical solvers—such as finite element, finite difference, and spectral methods—have established a foundation for computational science, recent advances in machine learning

have led to physics-informed neural networks (PINNs), which offer a flexible, data-driven approach to PDE approximation (Raissi et al., 2019a; Karniadakis et al., 2021).

Despite these advances, a central challenge remains: how can we balance expressivity, adaptivity, and efficiency to obtain accurate, tractable solutions for increasingly complex problems? The impact of these three factors differs fundamentally between classical and machine learning (ML) paradigms, creating complementary strengths and weaknesses. In this work, we decompose expressivity into three interrelated components: *function space richness*—the breadth and diversity of approximable functions; *adaptivity*—the ability to allocate expressive power selectively (whether across input regions or within the model's internal structure); and *efficiency per DoF*—the contribution of each parameter to the overall approximation. These criteria jointly determine a solver's ability to capture complex solution structures while maintaining computational tractability. The central motivation of this study is to combine the controllable adaptivity and efficiency of classical methods with the richness and flexibility of neural networks, unifying their advantages within a hybrid framework.

Classical numerical methods offer rigorous convergence guarantees and explicit, controllable adaptivity—allowing targeted mesh or basis refinement (Ainsworth & Oden, 1997). Such refinement is often guided by prior knowledge of solution smoothness or domain geometry (Brenner & Scott, 2008; Strang et al., 1974), leading to high efficiency per DoF, where each additional parameter meaningfully improves accuracy. However, their function space richness is constrained to the chosen basis or kernel, limiting the diversity of representable functions without significant model redesign. Adaptive extensions—such as *hp*-adaptive finite elements (Schwab, 1998) or greedy kernel selection (Schaback & Wendland, 2000)—can expand or restructure the function space, but operate through discrete, manually designed rules within a fixed basis family. In contrast, ML-based neural architectures adjust capacity continuously during training, reweighting and recombining learned features without altering the underlying discretization. Further, maintaining accuracy in complex scenarios, such as nonsmooth features, irregular geometries, or high-dimensional interactions, escalates the required DoFs, resulting in increased computational costs (Babuska & Suri, 1994). Thus, while classical methods excel in efficiency and controllable adaptivity, they are less inherently flexible in representing highly varied solution behaviors.

Conversely, PINNs offer a mesh-free, data-driven approach that embeds physical laws directly into the neural network training. PINN approaches are promising due to their rich function spaces, enabling the approximation of complex, high-dimensional problems without the need for hand-crafted bases (Hornik et al., 1989), and without requiring a priori knowledge of solution structure. The overparameterization of neural architectures enables this flexibility, but concurrently reduces the efficiency per DoF such that many parameters contribute marginally to the final accuracy (Adcock & Dexter, 2021). Unlike classical methods, where DoF efficiency is linked to convergence theory, its quantification in neural architectures remains largely empirical. Moreover, their adaptivity is largely implicit, governed by global optimization rather than targeted refinement, making it challenging to resolve localized features, sharp gradients, or multiscale phenomena (Wang et al., 2021b). While adaptive PINN methods exist, achieving architecture-level, localized adaptivity comparable to classical numerical methods remains an open challenge.

A growing body of research seeks to combine the complementary strengths of classical and ML paradigms for PDE approximation. One class enriches neural networks with fixed structured bases—such as Fourier, polynomial, or radial basis functions (RBFs)—to improve efficiency per DoF or representation quality (Cooley et al., 2025b; Wang et al., 2021b; 2023b). While some adaptive basis methods exist, such as trainable modes in random Fourier feature embeddings (Wang et al., 2021b), most basis-enrichment strategies fix the basis set in advance and introduce adaptivity through pruning or regularization. This imposes a predefined representational structure that the model can only reduce rather than expand. In contrast to the incremental refinement strategies of classical methods, this separation of adaptivity from expressivity can limit efficiency per DoF, particularly early in training, when many basis functions contribute little to the solution, or when pruning strategies are insufficient. Another class draws on mesh-refinement ideas, reallocating model effort toward difficult regions via adaptive sampling (Zeng et al., 2022), domain decomposition (Jagtap & Karniadakis, 2020), or architecture adjustments (Luo et al., 2025). These approaches reallocate expressive power locally—either by concentrating the training signal in regions of high residual (adaptive sampling) or by statically partitioning the domain so that subregions have dedicated parameters (domain decomposition and

mesh-inspired designs). As a result, function space richness remains unchanged, and adaptivity is achieved only indirectly through where and how the fixed capacity is applied—leaving a gap for methods that can dynamically expand and redistribute capacity while blending complementary function spaces throughout training.

We address this gap with HyResPINNs—a hybrid deep residual architecture that unifies function space richness, adaptivity, and efficiency per DoF in a single framework. Each residual block is a hybridization of two complementary function spaces: a trainable RBF neural network (RBFNN) that leverages compactly-supported RBFs for localized, structured refinement, and a deep neural network (DNN) for flexible, compositional expressivity. Their outputs are merged via a convex combination controlled by a trainable, interpretable gating parameter, allowing the model to adaptively shift between global and local modes over the course of training. A second gating parameter modulates the residual (skip) connection, starting at identity and progressively "opening" to activate additional depth only when needed, reducing early overparameterization. Stacking these hybrid blocks—that is, composing multiple blocks sequentially in depth—incrementally enriches the function space while concentrating refinement in specific solution regions, enabling fine-grained adaptivity and efficient parameter use—without reliance on static bases, coarse pruning, or fixed architectural capacity.

**Our key contributions are as follows:**

- **Expanded function space richness through hybridized composition.** We combine the global, compositional expressivity of deep neural networks with the localized approximation power of compactly-supported RBFs, enabling accurate representation of both smooth global structures and highly localized or discontinuous features across diverse PDE problems.

- **Fine-grained adaptivity via global–local blending and progressive depth activation.** Our architecture dynamically reallocates representational power during training—adjusting the balance between global and local modes and activating additional depth only when needed—to target refinement where it is most beneficial.

- **Improved efficiency per DoF through targeted capacity growth and localized enrichment.** HyResPINNs achieve competitive or superior accuracy with fewer, more effectively utilized parameters compared to static overparameterized neural PDE solvers by starting with minimal active capacity and selectively expanding it in response to solution complexity.

Through these contributions, we establish HyResPINNs as a practical approach for learning solutions to PDEs, providing a flexible framework that generalizes well across different problem domains. The remainder of this paper is organized as follows. In Section 2, we discuss related works and in Section 3, we review the relevant background. Then, in Section 4, we describe the mathematical formulation and architectural details of HyResPINNs. Next, in Section 5, we present numerical experiments comparing HyResPINNs to a suite of baseline neural PDE solvers, on a series of challenging PDE problems. We summarize our results and discuss future work in Section 6.

## 2 Related Work

In this section, we outline in more depth the approaches of classical numerical solvers and traditional PINNs. Then we review prior work in hybrid architectures that combine classical and ML paradigms, residual and stacked designs for progressive refinement, and radial basis function (RBF) methods and related kernel approaches. We organize this discussion to highlight how each class of methods addresses—or falls short of—the three critical components of expressive PDE solvers: (i) function space richness, (ii) adaptivity, and (iii) efficiency per degree of freedom. By situating HyResPINNs in relation to these prior works, we clarify both the limitations that motivate our framework and the distinct design principles that set it apart.

**Classical Numerical Solvers.** Classical numerical methods such as finite element (FEM) (Strang et al., 1974), finite difference (FDM) (LeVeque, 2007), and spectral approaches (Shen et al., 2011) have long provided a foundation for PDE solvers. These approaches employ carefully engineered discretizations or basis

functions informed by a priori knowledge of solution smoothness and domain geometry (Brenner & Scott, 2008; Strang et al., 1974). They offer provable convergence rates and explicit adaptivity, allowing refinement to be directly targeted to specific regions of complexity. For instance, they can concentrate additional DoFs in regions with singularities or sharp gradients by refining the approximation through mesh, basis, or combined strategies (Ainsworth & Oden, 1997; Melenk, 1997). However, the required DoF can escalate rapidly in the presence of nonsmooth features, irregular geometries, or high-dimensional interactions (Schwab, 1998; Babuska & Suri, 1994; Demkowicz et al., 2002; Bungartz & Griebel, 2004), resulting in significant computational costs.

**Physics-Informed Neural Networks (PINNs).** PINNs (Raissi et al., 2019a) approximate PDE solutions by incorporating physical laws into the neural network's loss function. PINNs achieve expressivity through learned, mesh-free representations, with function space richness emerging from network architecture, activation functions, and optimization processes (Hornik et al., 1989; Cybenko, 1989). Universal approximation theorems guarantee that, under certain architectural assumptions, neural networks can approximate any function arbitrarily well, suggesting their theoretical suitability for modeling complex PDEs. However, realizing this expressivity in practice is often difficult, with performance closely tied to the specific network architecture and optimization strategy (DeVore et al., 2020; Adcock & Dexter, 2021; Marwah et al., 2021). Standard PINNs exhibit implicit, global adaptivity: the network parameters are updated in response to the overall loss computed across the full domain, rather than through explicit, spatially localized refinement mechanisms typical in classical methods. Further, PINNs often perform poorly in regimes with steep gradients or multiscale features (Wang et al., 2022b). Recent approaches to ameliorating optimization issues include domain decomposition (X-PINNs) (Jagtap & Karniadakis, 2020), gradient-enhanced training (G-PINNs) (Yu et al., 2022), or discretely-trained PINNs using RBF-FD approximations in place of automatic differentiation (DT-PINNs) (Sharma & Shankar, 2022).

**Hybrid Architectures.** Efforts to hybridize PINNs with classical ideas often fall into two broad categories. The first are basis-enrichment methods which embed structured bases directly into neural architectures. Embedding known spectral or polynomial structure can yield high efficiency per DoF: each parameter has a targeted, interpretable role, and explicit representation of low- or high-frequency modes can offset the spectral bias of standard neural networks (Tancik et al., 2020; Cooley et al., 2025a). RBF-based methods, in contrast, target localized approximation and can provide improved computational efficiency when compactly supported kernels limit the region of influence (Wang et al., 2023b; Widrow & Lehr, 2002; Fasshauer, 2007). While some adaptive basis methods exist, such as trainable modes in random Fourier feature embeddings (Wang et al., 2021b), most basis-enrichment strategies fix the basis set in advance and introduce adaptivity through pruning or regularization. Across these methods, fixed, overcomplete basis sets are common, with adaptivity introduced only through pruning or regularization, which can decouple adaptivity from expressivity and limit efficiency per DoF when pruning is coarse.

Adaptivity-focused hybrids, in contrast, modify PINNs through adaptive sampling, domain decomposition, or architecture-level refinements inspired by mesh adaptation. These approaches reallocate expressive power locally—either by concentrating the training signal in regions of high residual (adaptive sampling) or by statically partitioning the domain so that subregions have dedicated parameters (domain decomposition and mesh-inspired designs). Recent efforts to introduce local adaptivity—including adaptive sampling (Zeng et al., 2022; Lu et al., 2021), loss reweighting (McClenny & Braga-Neto, 2020), and domain decomposition (Shukla et al., 2021; Jagtap & Karniadakis, 2020)—have improved performance in challenging regions, but these strategies still operate primarily at the data or loss level; the underlying parameter updates remain global, and truly local, mesh-like refinement within the network architecture remains largely unaddressed. Overall, most hybrids address only one axis of the expressivity–adaptivity–efficiency tradeoff at a time.

**Residual and Stacked Architectures.** Deep residual learning and stacked architectures have been recently introduced in scientific machine learning to overcome optimization challenges and enhance the expressivity of DNNs for solving PDEs. In particular, by enabling the training of deeper models and facilitating the learning of multi-scale phenomena through incremental refinement or additive corrections (He et al., 2016; Noorizadegan et al., 2024b). Residual connections allow deeper networks by alleviating gradient degrada-

tion, while stacked or multi-stage approaches iteratively refine solutions. These designs can increase function space richness without dramatically widening single models.

For instance, PirateNets (Wang et al., 2024) use multiple residual blocks with learned skip weights that modulate each block's influence. However, these residual weights are unconstrained and can grow arbitrarily large, complicating the interpretation of each block's non-linear transformation and potentially destabilizing training. Stacked PINNs (Howard et al., 2025) introduce a multifidelity paradigm in which the output of each trained network serves as a low-fidelity input for the next, iteratively building up model accuracy while reducing the need for excessively large single networks. This framework incrementally increases depth by stacking new PINNs, but it does so through a sequential training process that requires manual intervention at each stage. Both methods introduce partial adaptivity—PirateNets modulate block contributions; stacked PINNs increase depth over stages—but neither couples adaptivity, efficiency, and richness in a principled way. Crucially, neither integrates complementary function spaces (e.g., local RBFs with global DNNs) to exploit different inductive biases within a unified refinement framework.

**Radial Basis Function Networks and Kernel Methods.** Radial basis functions (RBF) methods have long been used in scientific computing as a flexible, mesh-free alternative to grid-based discretizations, especially for scattered or irregular domains (Wu et al., 2012; Fasshauer, 2007). Many RBFs—especially those with compact support or rapidly decaying influence—enable local control in function approximation, as each basis function primarily affects a specific region of the domain. This locality allows RBFs to accurately represent local solution features with minimal disturbance to other regions (Wu et al., 2012; Widrow & Lehr, 2002; Chen et al., 2023). As a result, RBFs are particularly advantageous in problems where local refinement is critical, whereas global bases such as Fourier or polynomial functions are often more effective for smooth, globally distributed features but can struggle near sharp transitions due to the Gibbs phenomenon.

Despite their strengths for localized approximation, traditional RBFNNs, particularly those with fixed centers, encounter the curse of dimensionality when approximating general smooth functions in high dimensions (Wu et al., 2012). The number of required basis functions can increase exponentially with input dimension. This contrasts with three-layer multilayer perceptrons, where the number of hidden units grows polynomially due to their global parameterization (Barron, 2002; Wu et al., 2012). This limitation highlights a broader challenge in approximating complicated functions: balancing local accuracy with efficient global representation. While RBFs offer locality, the field has also explored how other paradigms, such as neural networks and kernel methods, expand representational capacity. Specifically, theoretical and practical connections have been established between neural networks and kernel methods (Arora et al., 2019; Li et al., 2020a), inspiring feature lifting architectures such as random Fourier embeddings (Wang et al., 2021b; Li et al., 2020a) that aim to efficiently expand representational capacity. Crucially, while RBFNN-based models are valued for their mesh-free construction and strong locality, they generally lack the hierarchical, compositional structure and global parameter sharing that characterize DNNs—a key advantage shared by many feature lifting methods in expanding representational capacity efficiently. Yet most retain a bias toward either predominantly local (RBF-like) or predominantly global (DNN-like) representation.

**Positioning of HyResPINNs.** HyResPINNs directly address the limitations identified above by embedding trainable, compact-kernel RBFNNs and DNNs within a unified residual block architecture. Two interpretable gating parameters per block control (i) the convex combination between local (RBF) and global (DNN) modes, and (ii) the activation of the residual (skip) path, allowing progressive depth activation during training. This enables HyResPINNs to dynamically grow and redistribute capacity while blending complementary function spaces—coupling function space richness, adaptivity, and efficiency per DoF in a single architecture. Unlike prior hybrids, this approach jointly targets all three axes, enabling fine-grained, architecture-level refinement without manual basis selection, coarse pruning, or static overparameterization. HyResPINNs extend beyond prior residual or hybrid frameworks by introducing dual gating to simultaneously manage local–global feature blending and progressive depth activation. The inclusion of compactly-supported RBFs supplies explicit locality, addressing the lack of spatial adaptivity in residual and stacked architectures, while the residual gating mechanism ensures that model complexity grows smoothly with training.

**Operator Learning Frameworks.** *A parallel line of research learns mappings between function spaces rather than individual PDE solutions, including DeepONets (Goswami et al., 2023), Fourier Neural Operators (FNOs) (Li et al., 2020a), Graph Neural Operators (GNOs) (Li et al., 2020b), and GeoFNOs (Li et al., 2023). These models typically rely on data-driven training to learning operator mappings, which find solutions to a class of PDEs. In contrast, HyResPINNs solve single instances of PDEs; however, the architectural concepts introduced in this work could extend to the operator learning settings and represent and interesting future line of work on hybrid operator networks.*

## 3 Background

In this section, we review the core components underlying HyResPINNs: physics-informed neural networks (PINNs) as the baseline neural PDE framework, residual network architectures for progressive refinement, and radial basis function neural networks (RBFNNs) for localized approximation.

### 3.1 Physics-Informed Neural Networks.

Given a spatio-temporal domain $\mathcal{D} = [0, T] \times \Omega \subset \mathbb{R}^{1+d}$, where $\Omega$ is a bounded spatial domain in $\mathbb{R}^d$ with boundary $\partial\Omega$, the general form of a parabolic PDE is:

$$u_t + \mathcal{F}(u, \nabla u, \dots) = f, \tag{1}$$

where $\mathcal{F}$ is a linear or nonlinear differential operator, and $u$ denotes the unknown solution. This PDE is subject to an initial condition $u(0, \mathbf{x}) = g(\mathbf{x})$ for $\mathbf{x} \in \Omega$, and boundary conditions $\mathcal{B}(u(t, \mathbf{x})) = 0$ for $t \in [0, T]$ and $\mathbf{x} \in \partial\Omega$. Here, $f$ and $g$ are given functions, and $\mathcal{B}$ denotes an abstract boundary operator.

The general formulation of PINNs (Raissi et al., 2019a) aim to approximate the unknown solution $u(t, \mathbf{x})$ using a representation model $u_\theta(t, \mathbf{x})$, such that $u_\theta(t, \mathbf{x}) \approx u(t, \mathbf{x})$ and $\theta$ denotes the set of all trainable parameters of the network. The parameters $\theta$, are optimized to minimize the composite loss function:

$$L(\theta) = \lambda_{ic} L_{ic} + \lambda_{bc} L_{bc} + \lambda_r L_r, \tag{2}$$

where $L_{ic}$, $L_{bc}$, and $L_r$ represent the loss components associated with the initial conditions, boundary conditions, and the residual of the PDE, respectively; and $\lambda_{ic}$, $\lambda_{bc}$, and $\lambda_r$ are corresponding weighting coefficients that balance the contribution of each loss term. These terms can be set as static weights or adapted during training using various techniques such as NTK weighting (Wang et al., 2021b), gradient annealing (Wang et al., 2021a) among others.

Each loss term in Equation 2 is computed as the mean squared error of the initial condition, boundary, and PDE residuals. Specifically, each loss term is defined as:

$$L_{ic} = \frac{\lambda_{ic}}{N_{ic}} \sum_{i=1}^{N_{ic}} \left| u_\theta(0, \mathbf{x}_{ic}^i) - g(\mathbf{x}_{ic}^i) \right|^2, \tag{3}$$

$$L_{bc} = \frac{\lambda_{bc}}{N_{bc}} \sum_{i=1}^{N_{bc}} \left| \mathcal{B}[u_\theta](t_{bc}^i, \mathbf{x}_{bc}^i) \right|^2, \tag{4}$$

$$L_r = \frac{\lambda_r}{N_r} \sum_{i=1}^{N_r} \left| \frac{\partial u_\theta}{\partial t}(t_r^i, \mathbf{x}_r^i) + \mathcal{F}[u_\theta](t_r^i, \mathbf{x}_r^i) - f(t_r^i, \mathbf{x}_r^i) \right|^2. \tag{5}$$

The training data points $\{\mathbf{x}_{ic}^i\}_{i=1}^{N_{ic}}$, $\{t_{bc}^i, \mathbf{x}_{bc}^i\}_{i=1}^{N_{bc}}$ and $\{t_r^i, \mathbf{x}_r^i\}_{i=1}^{N_r}$ can be fixed mesh or points randomly sampled at each iteration of a gradient descent algorithm.

Variants of PINNs extend this basic formulation to improve stability, scalability, or compatibility with specific PDE structures. Specifically, Karniadakis and collaborators have extended these methods to conservative PINNs (cPINNs) (Jagtap et al., 2020), variational PINNs (vPINNS) (Kharazmi et al., 2019), parareal PINNs (pPINNs) (Meng et al., 2020), stochastic PINNs (sPINNs) (Zhang et al., 2019), fractional PINNs

(fPINNs) (Pang et al., 2018), LesPINNs (Yang et al., 2019), non-local PINNs (nPINNs) (Pang et al., 2020) and eXtended PINNs (xPINNs) (Jagtap & Karniadakis, 2020). While differing in implementation details, all variants rely on the ability of neural networks to represent complex solution spaces without manually designed bases, and all face the challenge of efficiently directing model capacity toward localized, high-complexity regions. In this work, we will focus on application of the original collocation PINNs approach; however, the work presented herein can be applied to many if not all of these variants.

## 3.2 Residual Networks

DNNs often encounter optimization challenges in deep architectures such as vanishing or exploding, resulting in training performance degradation. Residual Networks (ResNets) address these issues by introducing skip connections (He et al., 2016), which allow information to flow directly from an earlier layer to a later layer, creating an identity mapping that ensures information is preserved even as the network deepens (Noorizadegan et al., 2024a). Formally, a standard residual block is expressed as:

$$\mathbf{y} = \mathcal{F}(\mathbf{x}, \mathbf{W}_i) + \mathbf{x}, \tag{6}$$

where $\mathbf{x}$ represents the input to the block, $\mathcal{F}$ is the learned residual mapping, and $\mathbf{y}$ is the block output.

These connections improve gradient flow, enable stable training of deeper models, and allow the network to learn incremental refinements rather than full transformations from scratch (Noorizadegan et al., 2024a). In the PDE context, this incremental refinement aligns naturally with multi-scale and hierarchical solution structures, making residual architectures a useful foundation for adaptivity in depth. Recent work has explored variants with trainable skip weights, allowing the effective network depth to evolve during training, though often without constraints that ensure interpretability or stability (Wang et al., 2024).

| Name | Formula $\psi(r)$ | Support | Smoothness | Notes |
|------|-------------------|---------|------------|-------|
| Gaussian | $\exp\left(-\frac{1}{2}r^2\right)$ | Global | $C^\infty$ | $\sigma$ controls width |
| Multiquadric | $\sqrt{r^2 + c^2}$ | Global | $C^\infty$ | $c$ controls shape; non-decaying |
| Inverse MQ | $\frac{1}{\sqrt{r^2+c^2}}$ | Global | $C^\infty$ | Long-range; $c$ sets shape |
| Matérn-5/2 | $\left(1 + \sqrt{5}r/\ell + \frac{5}{3}(r/\ell)^2\right)e^{-\sqrt{5}r/\ell}$ | Global | $C^2$ | $\ell$ is the lengthscale |
| Wendland $C^2$ | $(1 - \frac{r}{\rho})^6(35(\frac{r}{\rho})^2 + 18(\frac{r}{\rho}) + 3),\ r < \rho$ | Compact | $C^2$ | $\rho$ sets support radius |

Table 1: Common radial basis functions $\psi(r)$ for RBFNNs. The distance $r(\mathbf{x}, \mathbf{x}^c; \omega_j)$ may be isotropic, anisotropic diagonal, or fully anisotropic (Mahalanobis), with $\omega_j$ denoting shape parameters (e.g., $\sigma$, $c$, $\ell$, $\rho$) for the $j$-th RBF.

## 3.3 Radial Basis Function Neural Networks

A Radial Basis Function Neural Network (RBFNN) is typically a shallow network with a single hidden layer of RBF units followed by a linear output layer. The output of the network is a linear combination of the kernel activations:

$$f(\mathbf{x}) = \sum_{j=1}^{N_c} c_j\,\psi(r(\mathbf{x}, \mathbf{x}_j^c;\,\omega_j)), \tag{7}$$

where $\mathbf{x}_j^c$ are center points, $c_j$ are coefficients, and $\psi$ is a chosen radial basis function kernel (see Table 1 for examples). The influence of each RBF is determined by a distance metric $r$ between the input $\mathbf{x}$ and the $j$-th center $\mathbf{x}_j^c$, with additional shape parameters $\omega_j$ (e.g., width or lengthscale). One metric is the isotropic Euclidean distance:

$$r = \frac{||\mathbf{x} - \mathbf{x}_j^c||}{\sigma_j}, \tag{8}$$

which applies a uniform scale in all directions. Anisotropic variants allow direction-dependent scaling, such as:

$$r = \sqrt{\sum_{i=1}^{d} \frac{(x_i - x_{j,i}^c)^2}{\sigma_{j,i}^2}}, \tag{9}$$

for axis-aligned ellipsoidal support (per-dimension widths $\sigma_{j,i}$), or more generally the Mahalanobis distance:

$$r = \sqrt{(\mathbf{x} - \mathbf{x}_j^c)^\top A_j (\mathbf{x} - \mathbf{x}_j^c)}, \tag{10}$$

using a symmetric positive-definite matrix $A_j$ for arbitrary orientation.

The specific choice of $\psi$ determines each basis function's support, smoothness, and decay—for example, Gaussians are globally supported and $C^\infty$ smooth, while Wendland functions are compactly-supported and less smooth. The shape parameters ($\sigma$, $c$, $A$, etc.) control locality and directional influence, and are often initialized to reasonable heuristics (e.g., based on data spread or uniformly), but can be further learned from data during training. RBFs with rapid decay or compact support intrinsically localize their influence, enabling accurate modeling of fine-scale features while minimizing impact on distant regions (Wu et al., 2012).

**Summary.** PINNs provide flexible, mesh-free function approximations; ResNets offer a mechanism for incremental refinement and stable training at depth; and RBFNNs give localized, interpretable basis functions well matched to targeted refinements. Together, these components form a foundation for architectures that can represent both global and local solution structures while controlling where and how model capacity is used.

## 4 Hybrid Residual PINNs (HyResPINNs)

We propose HyResPINNs, a deep learning framework that integrates radial basis function neural networks (RBFNNs) and deep neural networks (DNNs) within a hybrid residual architecture. This hybridization is designed to enhance function approximation for PDEs by combining the adaptive locality potential of RBFs with the expressive, hierarchical modeling capacity of DNNs.

### 4.1 Hybrid Residual Block: Dual Gating for Adaptive Representation

The core component of HyResPINNs is the hybrid residual block (see Figure 1), which combines two representational paradigms: an RBFNN component ($F_R$) and a DNN component ($F_N$). These are combined using two trainable gating parameters—one internal to each block and one external across blocks—to flexibly balance local and global contributions and to enable dynamic depth adaptation. We note that the degree of locality or globality realized in practice depends on the kernel choice and initialization strategies in the RBFNN component, and the chosen activation function in the DNN component.

#### 4.1.1 Internal Gating: Balancing Local and Global Features

Within each block, the outputs of the RBFNN and DNN components are merged via a trainable scalar parameter $\alpha^{(l)}$, transformed by a tempered sigmoid (Papernot et al., 2020) $\phi(z) = \phi_\tau(z/\tau)$, where $\tau > 0$ is a hyperparameter controlling gate sharpness and enforces the weights in $[0, 1]$. This ensures a convex combination:

$$F^{(l)}(\mathbf{x}) = \sigma\left( \phi(\alpha^{(l)}) F_R^{(l)}(\mathbf{x}) + \left(1 - \phi(\alpha^{(l)})\right) F_N^{(l)}(\mathbf{x}) \right), \tag{11}$$

where $F_R^{(l)}(\mathbf{x})$ and $F_N^{(l)}(\mathbf{x})$ are both projected to a common output dimension $p$, and $\sigma = \tanh$ ensures bounded, nonlinear transformations. Low values of $\tau$ yield smooth mixing; high values drives near binary selection between the component outputs. This mechanism enables the network to dynamically interpolate between localized adaptation provided by RBFs—especially with compact support or rapid decay—and global, compositional trends from DNNs.

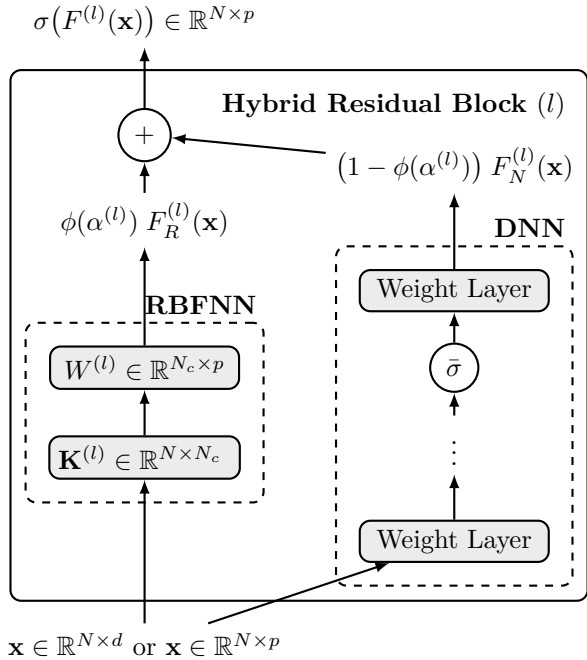

$$\sigma\big(F^{(l)}(\mathbf{x})\big) \in \mathbb{R}^{N \times p}$$

Figure 1: Illustration of the hybrid residual block ($l$), which combines contributions from RBFNN and DNN components. The input, $\mathbf{x} \in \mathbb{R}^{N \times d}$ for the first block ($\ell = 1$), or $\mathbf{x} \in \mathbb{R}^{N \times p}$ for inner blocks ($\ell > 1$), is first processed by the RBFNN through the kernel $\mathbf{K}^{(l)} \in \mathbb{R}^{N \times N_c}$, followed by a linear weight layer $W^{(l)} \in \mathbb{R}^{N_c \times p}$ to produce $F_R^{(l)}(x)$. In parallel, the DNN branch applies a sequence of weight layers and nonlinearities $\bar{\sigma}$ to yield $F_N^{(l)}(x)$. The two branches are adaptively fused through a trainable gating parameter $\alpha^{(l)}$, producing the weighted sum $\sigma\big(F^{(l)}(x)\big) = \phi(\alpha^{(l)})F_R^{(l)}(x) + \big(1 - \phi(\alpha^{(l)})\big)F_N^{(l)}(x)$, which is then passed forward through the residual connection. This design allows the model to balance kernel-based representations from the RBFNN with expressive features from the DNN, enhancing flexibility.

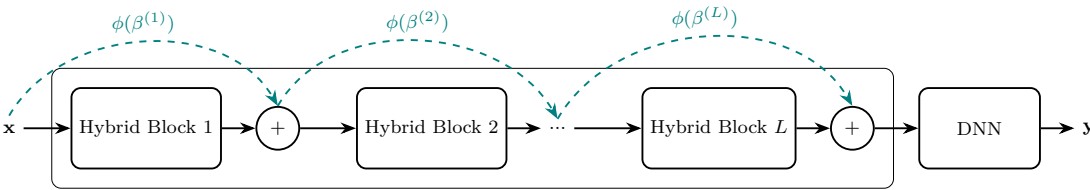

Figure 2: Illustration of the HyResPINN architecture using $L$ hybrid residual blocks. Each block combines outputs from an RBFNN component and a DNN component (see Figure 1). The trainable skip connections $\phi(\beta^{(\ell)})$, modulate the residual flow of information across blocks. These gated skip parameters enable the network to adaptively balance local kernel-based features and global neural representations at different depths, improving flexibility during training. The final output is obtained by passing the aggregated representation through a terminal DNN layer to produce the final prediction $\mathbf{y}$.

**The $p$ Parameter and Kernel Method Connection.** The block output dimension $p$ defines the feature lifting space where both components operate. This aligns with kernel methods, where data is explicitly mapped into a higher-dimensional feature space to enhance separability or approximation capabilities. As discussed in Section 2, theoretical and practical connections between neural networks and kernel methods inspire such architectures, aiming to efficiently expand representational capacity. In our design, the RBFNN and DNN components leverage this principle, enabling the subsequent layers to operate on more expressive features.

### 4.1.2 External Gating: Adaptive Residual Depth

A second gating parameter, $\beta^{(l)}$, is likewise passed through a temperature-controlled sigmoid to modulate the residual connection:

$$H^{(l)} = \phi(\beta^{(l)})F^{(l)} + (1 - \phi(\beta^{(l)}))H^{(l-1)}, \tag{12}$$

where $H^{(l-1)}$ is the previous block's output (or the original input for $l = 1$). This adaptive gating enables the network to learn how much each block should transform the representation, thus supporting dynamic depth during training.

### 4.1.3 Regularization and Initialization

We regularize gating parameters to avoid over-reliance on a single pathway similar to (Howard et al., 2025):

$$\mathcal{L} = \lambda_{ic}\mathcal{L}_{ic} + \lambda_{bc}\mathcal{L}_{bc} + \lambda_r\mathcal{L}_r + \lambda_g \sum_{i=1}^{L} \left(\alpha_i^2 + \beta_i^2\right), \tag{13}$$

such that $\lambda_g$ is a weight modulating the strength of the regularization.

We initialize $\phi(\beta^{(l)}) = 0$ for all $l$, biasing the network toward the residual (identity) mapping at the start (Hauser, 2019), and $\phi(\alpha^{(l)}) = 0.5$ to initially balance RBFNN and DNN contributions. DNN weights and RBF coefficients use Xavier (Glorot & Bengio, 2010) initialization, while the RBF shape parameters are initialized using a quantile distance heuristic between the center points in each block. The input center points are initialized using the Poisson sampling method of (Shankar, 2017), and internal block center points are initialized using k-means clustering of the subsequent block's output following (Wurzberger & Schwenker, 2024).

## 4.2 RBFNN Block Component

For input $\mathbf{x} \in \mathbb{R}^{N \times d}$, the $l$-th RBFNN constructs:

$$\mathbf{K}_{ij}^{(l)} = \psi^{(l)}\left(r\left(\mathbf{x}_i, \mathbf{x}_j^c; \omega_j^{(l)}\right)\right). \tag{14}$$

Here, $\psi^{(l)}$ denotes the selected RBF (see Table 1), and $r(\cdot, \cdot; \omega_j^{(l)})$ is a distance metric parameterized by trainable shape parameters $\omega_j^{(l)}$, which control the width or anisotropy of each basis. The kernel matrix is multiplied by a trainable weight matrix $W^{(l)} \in \mathbb{R}^{N_c \times p}$, yielding the final RBFNN output:

$$F_R^{(l)}(\mathbf{x}) = \mathbf{K}^{(l)}(\mathbf{x}) \cdot W^{(l)} \in \mathbb{R}^{N \times p}. \tag{15}$$

Each hybrid block maintains its own set of RBF center points $\{\mathbf{x}_j^c\}^{(l)}$, allowing block-specific local refinements that adapt independently across network depth.

**Wendland $C^2$ Kernel Selection.** In this work, we primarily use the $C^2(\mathbb{R}^3)$ Wendland kernel defined as:

$$\psi_{\text{Wend},C^2}(r) = (1 - r)_+^6 (35r^2 + 18r + 3), \tag{16}$$

where $(\cdot)_+ = \max\{0, \cdot\}$ ensures compact support and $\mathbf{x}_i^c$ is the center of the $i$-th RBF. The variable $r$ denotes the distance metric, which may be either isotropic (Equation 8) or anisotropic (Equation 9 or Equation 10). We implement the Wendland kernels with unit support ($\rho = 1$ from Table 1), such that the kernel evaluates to zero whenever $r \geq 1$, ensuring strict locality and sparse kernel matrices. Fixing $\rho = 1$ lets the anisotropy control the effective radii directly, simplifying the parameterization without sacrificing expressivity. For the fully anisotropic case in Equation 10, we parameterize each matrix $A_j$ in Cholesky form, $A_j = M_j^\top M_j$, where $M_j$ is a trainable lower-triangular matrix with positive diagonal entries. This guarantees that $A_j$ remains symmetric positive-definite during training, which ensures well-defined distances, avoids degeneracies, and is practically efficient (Wurzberger & Schwenker, 2024). Empirically, ablation studies confirmed that the $C^2$ Wendland RBF achieves the lowest approximation error compared to alternative kernels, justifying its use in our architecture.

### 4.3 DNN Block Component

Each DNN block component computes:

$$F_N^{(l)}(\mathbf{x}) = \mathcal{NN}(\mathbf{x}; \theta^{(l)}) \in \mathbb{R}^{N \times p}, \tag{17}$$

where $\theta^{(l)}$ represents the set of trainable parameters in the network. The final output is projected to the $p$-dimensional feature space shared by the RBFNN component. By design, this block can use any standard or advanced DNN architecture; in some experiments, we employ Modulated Multi-Layer Perceptrons (ModMLPs) with random weight factorization (Wang et al., 2021a), which are an extension to standard fully connected networks inspired by neural attention mechanisms. These modifications incur relatively small computational and memory overhead while leading to significant improvements in predictive accuracy (Wang et al., 2023a).

### 4.4 Full Model Composition

The complete HyResPINN model (see Figure 2) consists of $L$ stacked hybrid residual blocks. After the final block, a standard DNN head further refines the representation into the final prediction:

$$u_\theta = \mathrm{DNN}_{out}(H^{(L)}). \tag{18}$$

In some examples, we initialize the final layer of the final DNN using the physics-informed approach of (Wang et al., 2024), a least-squares fit to the PDE residuals at collocation points before gradient training which embeds physics priors and accelerating convergence.

**Summary.** By fusing RBFNNs with DNNs in a residual, gated architecture, HyResPINNs achieve function space richness, fine-grained adaptivity, and improved parameter efficiency. Our experiments demonstrate that this approach generalizes robustly across canonical and irregular PDE benchmarks, outperforming state-of-the-art neural PDE solvers in both accuracy and parameter efficiency.

| Problem | PINN | ResPINN | ExpertPINN | StackedPINN | PirateNet | **HyResPINN** |
|---|---|---|---|---|---|---|
| Allen–Cahn (Dirchlet BCs) | 2.65e-1 | 6.82e-2 | 9.66e-3 | 6.11e-2 | 1.10e-2 | **2.39e-3** |
| Allen–Cahn (Periodic BCs) | 5.26e-1 | 2.70e-3 | 3.86e-3 | 5.87e-3 | 2.62e-5 | **1.19e-5** |
| DarcyFlow (2D Dirichlet BCs) | 7.50e-4 | 4.60e-4 | 5.00e-4 | 9.00e-4 | 8.71e-5 | **5.44e-5** |
| (2D Neumann BCs) | 2.00e-3 | 1.40e-3 | 1.20e-4 | 4.10e-3 | 1.70e-4 | **6.00e-5** |
| (3D Dirichlet BCs) | 2.20e-3 | 1.20e-2 | 2.10e-2 | 5.40e-2 | 3.90e-2 | **1.20e-3** |
| (3D Neumann BCs) | 8.50e-3 | 6.10e-3 | **1.10e-3** | 3.90e-2 | 1.30e-3 | **1.10e-3** |
| DarcyFlow (rough) | 6.85e-5 | 2.69e-5 | 1.10e-4 | 1.50e-4 | 5.44e-5 | **1.05e-5** |
| Kuramoto–Sivashinsky | 1.42e-1 | 1.29e-2 | 1.57e-3 | 1.86e-1 | 2.00e-3 | **8.12e-4** |
| Korteweg–De Vries | 8.17e-1 | 5.72e-1 | 6.86e-4 | 2.98e-1 | 5.61e-4 | **4.09e-4** |
| Grey–Scott ($u$) | 7.54e-1 | 9.87e-3 | 4.79e-3 | **1.21e-3** | 3.61e-3 | 6.43e-3 |
| Grey–Scott ($v$) | 9.56e-1 | 1.10e-2 | **8.98e-3** | 6.90e-2 | 9.39e-3 | 1.31e-2 |

Table 2: Relative $\ell_2$ test error results across a range of PDE benchmarks under different boundary conditions. For the Grey–Scott PDE, errors are reported separately for both components ($u$ and $v$). As a summary table, the lowest error for each problem (i.e., each row) is highlighted in bold and second-lowest errors are underlined. Note that in some cases, competing methods arrive at similar results.

## 5 Experimental Results and Discussion

We evaluate HyResPINNs on a suite of PDE benchmark problems designed to test its function space richness, adaptivity, and efficiency per DoF. These benchmarks have been selected to either model solution features that present significant challenges for computational models or are recognized as established benchmark problems. First, to evaluate localized adaptivity, we consider two 1D non-linear hyperbolic Allen-Cahn

equations, one with Dirichlet boundary conditions in Section 5.1, and periodic boundary conditions in Section 5.2, each containing sharp solution features. In Section 5.3, we test 2D and 3D Darcy flow problems on annulus and extruded annulus domains to evaluate generalization to non-standard geometries using various boundary conditions. Then, Section 5.4 details a 2D Darcy flow problem with rough, discontinuous coefficients to examine robustness under discontinuities. To further explore the function space richness of our methods, we evaluate the chaotic and multi-scale systems given by the Kuramoto–Sivashinsky equation in Section 5.5, the Korteweg-de Vries equation in Sections 5.6, and the Grey–Scott equation in Section 5.7. We assess predictive accuracy and efficiency per DoF throughout all experiments, comparing against state-of-the-art baselines. Table 2 summarizes the top results from each test.

**Baseline Methods** We compare HyResPINNs against a suite of baseline methods. These include 1) the standard PINN as originally formulated in (Raissi et al., 2019b) (PINN); 2) a PINN based on the architectural guidelines proposed in (Wang et al., 2023a) (ExpertPINNs); 3) PINNs with residual connections (ResPINNs); 4) PirateNets (Wang et al., 2024); and 5) the stacked PINN approach of (Howard et al., 2025) (StackedPINNs). This baseline suite was selected to capture the incremental steps leading to the proposed architecture. Each comparison isolates one primary architectural mechanism—ResPINNs isolate residual skip connections; ExpertPINNs isolate Fourier feature embeddings and high-dimensional residual skip connections; PirateNets isolate adaptive skip connections within a block structure; StackedPINNs isolate convex combinations between two types of approximators. Collectively, these baselines incrementally probe the three axes of expressivity, adaptivity, and efficiency per DoF. Standard PINNs rely on a fixed DNN representation with limited expressivity and no adaptive structure. ResPINNs extend this baseline by incorporating residual skip connections, improving training stability and adaptive depth utilization, while moderately improving efficiency. ExpertPINNs enhance expressivity through Fourier feature embeddings, while PirateNets focus on adaptivity by learning dynamic skip connections that regulate residual information flow. StackedPINNs increase expressivity through convex combinations of multiple sub-networks. Together, these methods progressively explore the expressivity-adaptivity-efficiency space that HyResPINNs unify within a single architecture—combining local and global representations through dual learnable gates. We implemented each approach following the architectural details provided to ensure a fair comparison.

**Experimental Setup** We follow similar experimental design procedures as those described in (Wang et al., 2022a; 2023a; 2024). We employ mini-batch gradient descent for most experiments, with collocation points randomly sampled during each training iteration. In the Darcy Flow experiments, we use full-batch gradient descent, with collocation point sets generated using the efficient Poisson sampling technique from (Shankar, 2017). For training, we use the Adam optimizer Kingma & Ba (2015), and follow the learning rate schedule of (Wang et al., 2023a) which starts with a linear warm-up phase of $5,000$ iterations, starting from zero and gradually increasing to $10^{-3}$, followed by an exponential decay at a rate of 0.9. We also employ a learning rate annealing algorithm (Wang et al., 2023a) to balance each loss term and causal training (Wang et al., 2022a; 2023a) to mitigate causality violation in time-dependent PDEs. Further, we apply exact periodic or Dirichlet boundary conditions (Dong & Ni, 2021) when applicable. We use the hyperbolic tangent activation functions in all DNNs and initialize each network's parameters using the Glorot normal scheme (Glorot & Bengio, 2010). We ran five random trials for each test and report the mean values achieved in each plot and table. The code for our methods and all compared baseline approaches is written in Jax $0.6.0$[1]. We train all models on an Nvidia H100 GPU running CentOS 7.2.

**Comparison Metrics** We measure the quality of each model's predicted solution $u_\theta$ using the relative $\ell_2$ error, defined as $\mathcal{E}_2 = \frac{\|u_\theta - u^*\|_2}{\|u^*\|_2}$, where $u^*$ denotes the reference solution (either exact or computed using a high-fidelity solver). Here $\|\cdot\|_2$ refers to the discrete $\ell_2$ norm evaluated over a set of equispaced test points in the domain, which serves as an approximation of the continuous $L^2(\Omega)$ norm. We quantify model efficiency as a function of accuracy, parameter count, and computational cost. Formally, we define the efficiency per DoF (denoted $\eta_{\mathrm{DoF}}$) as:

$$\eta_{\mathrm{DoF}} = \frac{1}{\mathcal{E}_2 \cdot N_{\mathrm{params}} \cdot T_{\mathrm{train}}}, \tag{19}$$

---

[1] https://github.com/madicooley/HyResPINNs

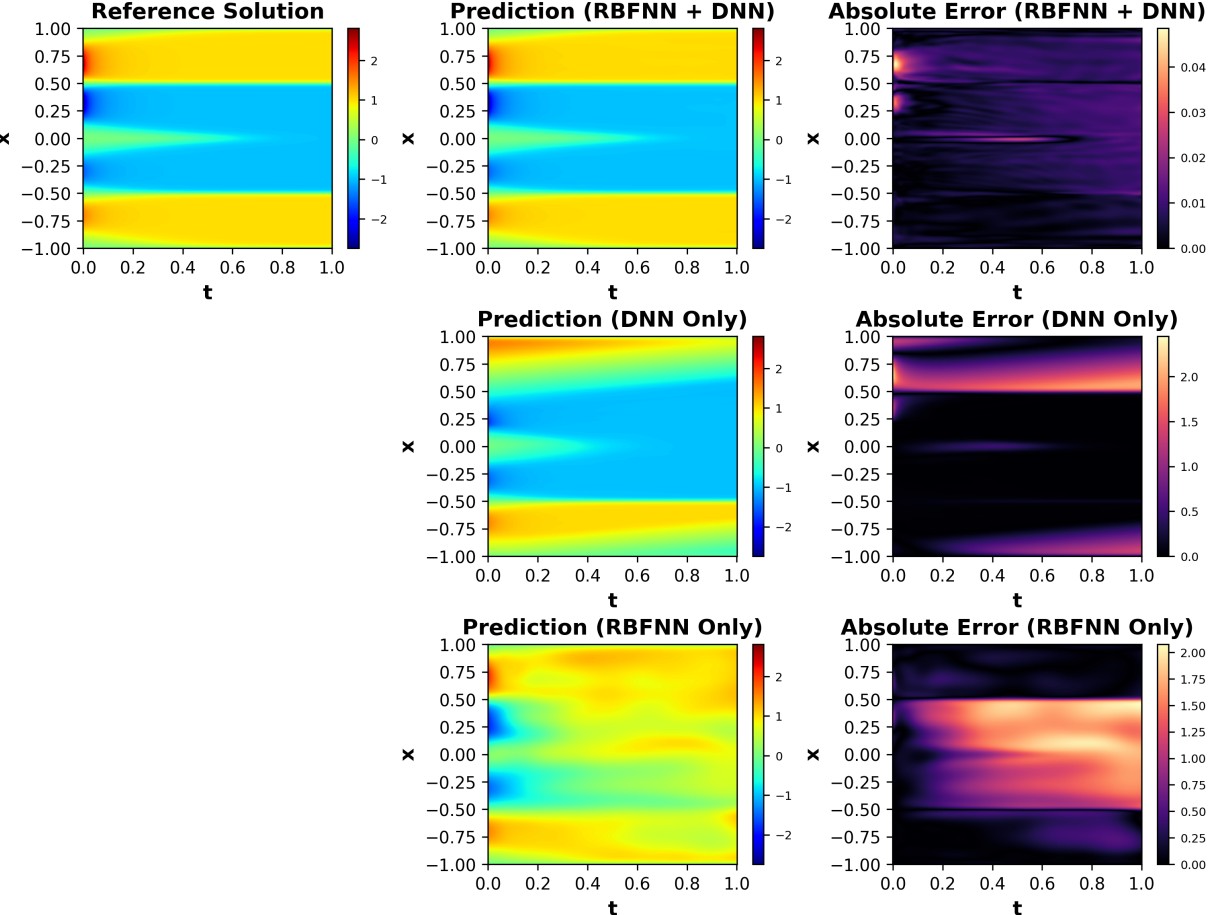

(a) Comparison of the three variants: HyResPINN (RBFNN+DNN), DNN-only, and RBFNN-only's predicted solutions (middle) and the absolute errors (right) compared to the true solution (left).

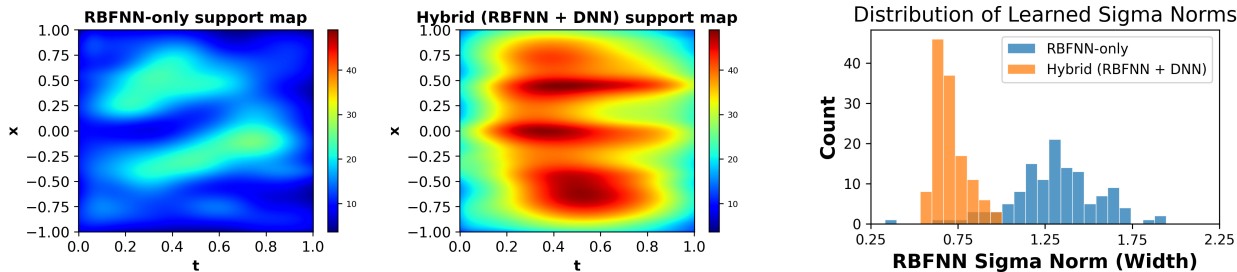

(b) (Left) RBF Support Map. Hybrid model exhibits stronger kernel coverage at sharp solution features. (Right) Sigma Norms. Distribution of learned RBF widths. Hybrid learns broader, smoother kernels.

Figure 3: *Allen–Cahn equation (Dirichlet BCs):* Comparison of model outputs and internal representations across variants.

where $N_{\text{params}}$ is the number of trainable parameters (the model's degrees of freedom), and $T_{\text{train}}$ represents the mean computational cost of training, measured as the average number of epochs completed per second under identical hardware and batch configurations. Intuitively, a model with higher $\eta_{\text{DoF}}$ achieves more accurate solutions relative to its complexity and computational cost, resulting in higher efficiency per DoF.

To further analyze efficiency per DoF, we decompose $\eta_{\text{DoF}}$ into two complementary metrics that isolate different aspects of model performance:

$$\eta_{\text{params}} = \frac{1}{\mathcal{E}_2 \cdot N_{\text{params}}}, \quad \eta_{\text{comp}} = \frac{1}{\mathcal{E}_2 \cdot T_{\text{train}}}. \tag{20}$$

Here, $\eta_{\text{params}}$ (parameter efficiency) quantifies the accuracy achieved per degree of freedom. Conversely, $\eta_{\text{comp}}$ (computational efficiency) measures accuracy per unit training time, indicating how effectively the model architecture and optimization routine utilize computational resources.

Intuitively, a model can minimize the error $\mathcal{E}_2$ to an arbitrary extent by increasing the number of trainable parameters $N_{params}$ or extending training time $T_{train}$. However, doing so diminishes its overall efficiency. In practice, two models with identical accuracy can exhibit markedly different efficiency values: one achieving low error compactly and quickly, the other reaching the same accuracy only through overparameterization or prolonged optimization. Thus, high accuracy alone does not imply superior performance—a model that achieves slightly higher error yet trains faster or with fewer degrees of freedom may, in effect, be more efficient. Together, these fine-grained metrics provide a comprehensive view of efficiency per DoF, clarifying how different design choices balance expressivity, computational cost, and solution accuracy.

### 5.1 1D Allen-Cahn with Dirichlet Boundary Conditions

We start by considering the one-dimensional Allen–Cahn equation defined on $x \in [-1, 1]$, $t \in [0, 1]$ given as:

$$\frac{\partial u}{\partial t} = \varepsilon \frac{\partial^2 u}{\partial x^2} - 5u^3 + 5u,$$

where $u(x, t)$ is the solution and $\varepsilon = 0.001$ is the diffusion coefficient. We impose the homogeneous Dirichlet boundary conditions:

$$u(-1, t) = 0, \quad u(1, t) = 0, \quad \text{for all } t \in [0, 1],$$

and the initial condition is:

$$u(x, 0) = -2\sin(2\pi x)\left(2e^{-8(x-0.5)^2} - e^{-5(x+0.5)^2}\right).$$

This nonlinear, reaction-diffusion PDE's solution exhibits both smooth, global variation and sharp, localized peaks (centered at $x = \pm 0.5$). This problem is designed to evaluate a model's capacity to accurately capture these localized features while simultaneously preserving the global solution structure.

We first perform ablation studies to systematically investigate the individual contributions of the RBFNN and DNN components within the HyResPINN residual blocks. Next, to measure the DoF efficiency, we benchmark the complete HyResPINN model against baseline solvers with comparable model complexity, quantified by parameter count and network depth. Appendix A details ablation studies measuring the accuracy and computational cost of the HyResPINN architecture under varying RBF kernel types, numbers of center points, and deepening residual blocks. For the remainder of the experiments, we fix the RBF kernel to Wendland $C^2$—the configuration our ablation studies identify as the most effective concerning accuracy, computational efficiency, while aligning with our goal of inherent locality.

*Expressivity: Local and Global Representations.* We first examine the function space richness of HyResPINNs by isolating local (RBFNN), global (DNN), and hybrid (RBFNN+DNN) representations, testing whether their combination enables more expressive solution approximation than either alone. To isolate the contributions of each hybrid component, we conduct ablation experiments comparing RBFNN-only, DNN-only, and full HyResPINN configurations. In the RBFNN-only configuration, the DNN outputs are fixed to zero, isolating localized basis function representation. In the DNN-only configuration, the RBFNN outputs are

fixed to zero, isolating global neural representation. In the full HyResPINN, both components are active and combined via learned gating. All models are trained with Adam for 100,000 iterations, with a batch size of 1024 collocation points randomly sample each iteration. The DNNs use three layers of 64 neurons, RBFNNs use 64 centers, and HyResPINNs use both and one block.

As shown in Figure 3a, the full hybrid model achieves the lowest relative error ($2.4e-3$), accurately resolving both the steep gradients and the global structure. The DNN-only baseline captures the overall shape but fails to resolve sharp peaks, resulting in localized errors above 1.0 (relative error $2.65e-1$). The RBFNN-only baseline captures localized peaks but struggles to maintain smoothness elsewhere ($4.6e-3$). Neither baseline achieves the accuracy or robustness of the hybrid. These results directly confirm that hybrid local–global representation enables superior function approximation on challenging, multiscale PDEs.

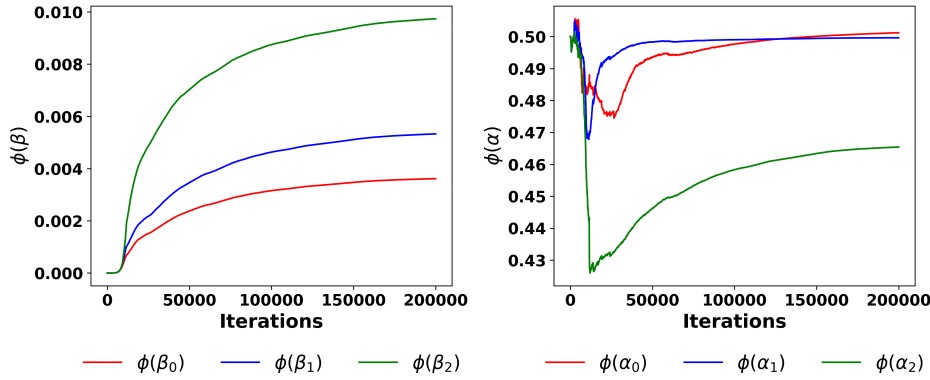

Figure 4: *Allen–Cahn equation (Dirichlet BCs):* Evolution of the learned gating parameters during training. (Left) Evolution of residual gating parameters $\phi(\beta_i)$, which modulate the strength of residual information flow across blocks. (Right) Evolution of hybrid gating parameters $\phi(\alpha_i)$, which balance the DNN and RBF components within each block.

*Adaptivity: Solution Refinement.* We next examine how HyResPINNs achieve fine-grained adaptivity in the width of RBF kernels. Kernel support maps (Figure 3b, left), computed by summing RBF activations across the domain, reveal that the hybrid model develops broad, overlapping support, densely covering regions with sharp features. In contrast, the RBFNN-only model forms narrowly focused kernels with little overlap. Furthermore, the distribution of learned RBF widths (Figure 3b, right) demonstrates that HyResPINNs favor wider, smoother kernels compared to RBFNNs alone. This broader support reflects the model's ability to blend basis functions, promoting both smooth transitions and precise local refinement—key ingredients of adaptive expressivity.

Additionally, we conducted an ablation study analyzing the impacts of different parameter initialization strategies; we report this in Appendix A.3. Overall, the results indicated that the model is largely robust to initializations of both parameters apart from when $\phi(\alpha)$ is initialized to 1 and $\phi(\beta)$ is initialized to 1. Therefore, we opt to initialize $\phi(\beta)$ to 0 such that the start is the identity and initialize $\phi(\alpha)$ to 0.5 so the components start with equal contribution. Figure 4 shows the evolution of the gating parameters during training. The $\phi(\beta)$ gating parameters increase steadily during training, indicating the progressive strengthening of residual information flow across the blocks. The hybrid gating parameters $\phi(\alpha)$ initially decrease—reflecting a stronger reliance on the RBFNN components—before stabilizing at values that indicate a moderate preference for the RBFNN.

*Efficiency: Accuracy and Convergence per Degree of Freedom.* We evaluate the empirical efficiency of HyResPINNs relative to competing PINN baselines under comparable parameter budgets, focusing on both the error convergence behavior and overall accuracy. As shown in Figure 5, HyResPINNs achieve lower mean relative $\ell_2$ error and faster convergence than the baseline models. Table 3 further quantifies these trends using the empirical efficiency metrics defined in Equation 19 and Equation 20. Although HyResPINNs do not exhibit the lowest computational cost (training fewer iterations per second than some alternatives), they achieve the highest parameter efficiency ($\eta_{\text{params}} = 2.358e-2$), and highest computational efficiency

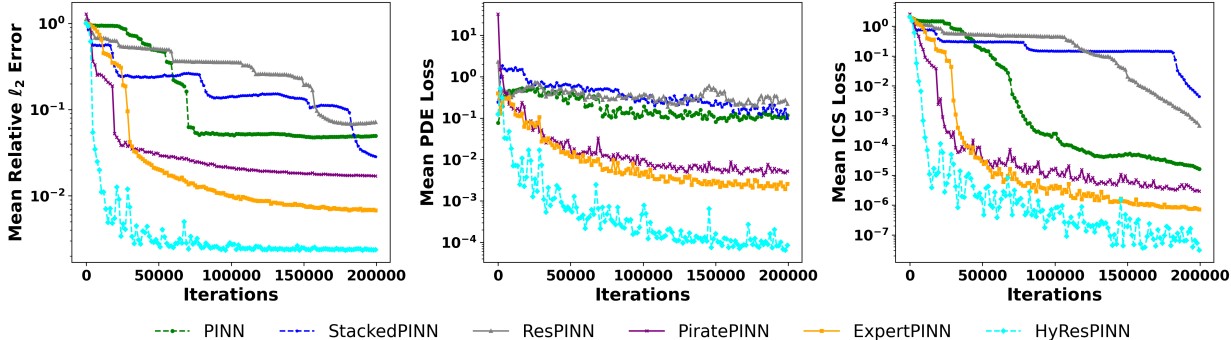

Figure 5: *Allen–Cahn equation (Dirichlet BCs):* (Left) Mean relative $\ell_2$ errors, (middle) PDE loss, (right) initial condition loss, across all baselines of similar parameter numbers as a function of training iteration.

| Model | $\mathbf{N}_{\mathrm{params}}$ | $\mathbf{T}_{\mathrm{train}}$ | Rel $\ell_2$ Error | $\eta_{\mathrm{params}}$ | $\eta_{\mathrm{comp}}$ | $\eta_{\mathrm{DoF}}$ |
|---|---|---|---|---|---|---|
| PINN | 17,025 | 8264 (232) | 4.98e-02 (0.00986) | 1.179e-3 | 2.430e-3 | 1.427e-7 |
| StackedPINN | 17,359 | 5705 (7) | 2.83e-02 (0.04217) | 2.036e-3 | 6.194e-3 | 3.568e-7 |
| ResPINN | 17,157 | 6734 (136) | 7.18e-02 (0.03379) | 8.118e-4 | 2.068e-3 | 1.205e-7 |
| PiratePINN | 17,926 | 4623 (26) | 1.70e-02 (0.00848) | 3.281e-3 | 1.272e-2 | 7.098e-7 |
| ExpertPINN | 17,793 | 5394 (52) | 6.80e-03 (0.00025) | 8.265e-3 | 2.726e-2 | 1.532e-6 |
| HyResPINN | 17,667 | 5549 (9) | 2.40e-03 (0.00021) | 2.358e-2 | 7.508e-2 | 4.250e-6 |

Table 3: *Allen–Cahn equation (Dirichlet BCs):* Comparison of model complexity and efficiency across PINN variants. Reported are the total number of trainable parameters ($N_{\mathrm{params}}$); the computational cost, expressed as the average number of training iterations completed per second ($T_{\mathrm{train}}$) with standard deviation in parentheses; the mean relative $\ell_2$ error with standard deviation in parentheses; and the corresponding efficiency metrics $\eta_{\mathrm{params}}$, $\eta_{\mathrm{comp}}$, and $\eta_{\mathrm{DoF}}$. Higher efficiency values indicate models that achieve lower relatives errors relative to their representational and computational costs.

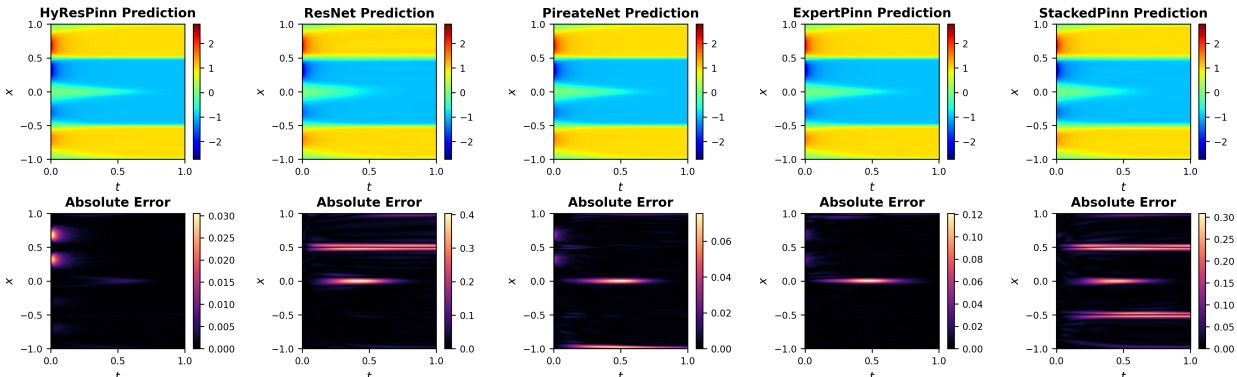

Figure 6: *Allen–Cahn equation (Dirichlet BCs):* Predicted solutions (top row) and corresponding absolute errors (bottom row) for various baseline models. Each column shows the model's prediction across the full domain, followed by the absolute errors relative to the reference solution.

($\eta_{\text{comp}} = 7.508e-2$), resulting in the highest overall composite efficiency ($\eta_{\text{DoF}} = 4.250e-6$). This indicates that HyResPINNs attain lower errors relative to their representational complexity and runtime cost. Qualitative comparisons in Figure 6 further show that only HyResPINNs robustly capture both sharp and smooth solution features, whereas most baseline methods fail to recover the sharp solution features, exhibiting large absolute errors near $x = \pm 0.5$.

*Summary.* These analyses highlight the benefit of hybridization: the RBFNN component provides localized adaptivity, while the DNN component ensures global coherence. HyResPINN's lower error and more adaptive internal structure demonstrate that coordinated local–global representation is essential for robust and expressive PDE learning.

### 5.2 1D Allen-Cahn with Periodic Boundary Conditions

Next, we present results for 1D the Allen-Cahn problem with periodic boundary conditions, a well-known challenge for conventional PINN models and a widely studied benchmark in recent literature (Wight & Zhao, 2020; Wang et al., 2022a; Daw et al., 2022). For simplicity, we consider the one-dimensional case with a periodic boundary condition with $t \in [0, 1]$, and $x \in [-1, 1]$:

$$u_t - 0.0001 u_{xx} + 5u^3 - 5u = 0\,,$$
$$u(0, x) = x^2 \cos(\pi x)\,,$$
$$u(t, -1) = u(t, 1)\,, \ u_x(t, -1) = u_x(t, 1)\,.$$

This problem is particularly difficult for two reasons. First, the Allen-Cahn PDE requires PINNs to approximate solutions with sharp transitions in space and time, posing challenges for standard neural network architectures. Second, there is an inherent incompatibility between the initial and boundary conditions, leading to a weak discontinuity at time $t = 0$. Traditional numerical methods would likely produce oscillations near the discontinuity due to the mismatch between the initial and boundary conditions. However, unlike traditional solvers, PINNs do not rely on discretization and instead learn their model parameters through optimization. This optimization process may mitigate spurious oscillations, as it relies on stochastic updates and continuous function approximations. Specifically, implicit regularization from both mini-batch training and stochastic gradient descent (SGD) may contribute to stabilizing optimization and reducing any spurious oscillations (Sekhari et al., 2021). Full experimental details are provided in Appendix B.

*Expressivity: Capturing Periodic and Discontinuous Solutions.* Similar to the previous section, we compare HyResPINNs to DNN-only and RBFNN-only variants. Figure 7 demonstrates the advantages of the hybrid residual block structure in capturing both smooth and sharp features in the 1D Allen-Cahn solution. The absolute error plots show that HyResPINN exhibits significantly lower errors than standard DNN-only and RBFNN-only, validating its improved accuracy.

| Model | $N_{\text{params}}$ | $T_{\text{train}}$ | Rel $\ell_2$ Error | $\eta_{\text{params}}$ | $\eta_{\text{comp}}$ | $\eta_{\text{DoF}}$ |
|---|---|---|---|---|---|---|
| PINN | 724,353 | 1947 (6) | 4.54e-03 (0.00010) | 3.038e-4 | 1.130e-1 | 1.560e-7 |
| ExpertPINN | 727,170 | 1309 (10) | 3.86e-05 (0.00004) | 3.563e-2 | 1.979e1 | 2.721e-5 |
| PiratePINN | 727,171 | 1355 (4) | 2.62e-05 (0.00001) | 5.249e-2 | 2.816e1 | 3.873e-5 |
| ResPINN | 790,145 | 2114 (11) | 1.53e-03 (0.00016) | 8.285e-4 | 3.096e-1 | 3.919e-7 |
| StackedPINN | 793,738 | 833 (3) | 5.87e-03 (0.00012) | 2.146e-4 | 2.044e-1 | 2.576e-7 |
| HyResPINN | 759,686 | 827 (2) | 1.19e-05 (0.00001) | 1.106e-1 | 1.016e2 | 1.337e-4 |

Table 4: *Allen–Cahn equation (Periodic BCs):* Comparison of model complexity and efficiency across PINN variants. Reported are the total number of trainable parameters ($N_{\text{params}}$); the computational cost, expressed as the average number of training iterations completed per second ($T_{\text{train}}$) with standard deviation in parentheses; the mean relative $\ell_2$ error with standard deviation in parentheses; and the corresponding efficiency metrics $\eta_{\text{params}}$, $\eta_{\text{comp}}$, and $\eta_{\text{DoF}}$. Higher efficiency values indicate models that achieve lower relatives errors relative to their representational and computational costs.

*Adaptivity: Depth Ablation and Solution Refinement.* To test architectural adaptivity, we vary the number of residual blocks. Figure 8 demonstrates that increasing the number of residual blocks in HyResPINNs leads to

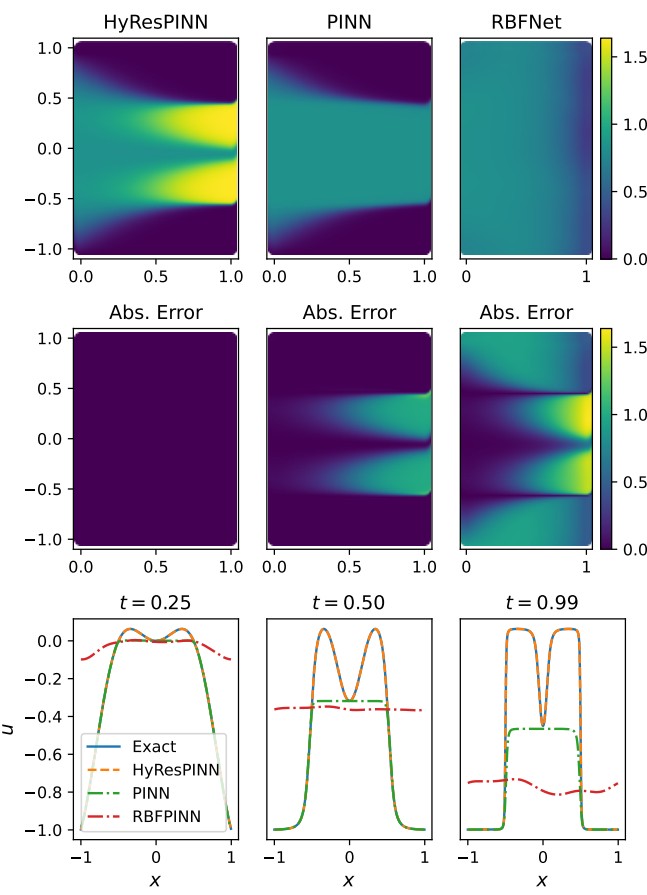

Figure 7: *Allen-Cahn equation (Periodic BCs):* Comparison of the predicted solutions for the Allen-Cahn equation using HyResPINN, standard PINN, and RBF PINN models. The top row shows the predicted solutions for HyResPINN (left), standard PINN (center), and RBF PINN (right). The second row shows the absolute error between the predicted and true solutions. The bottom row shows the predicted solutions for time steps ($t = 0.25, 0.5, 0.99$) compared to the true solution.

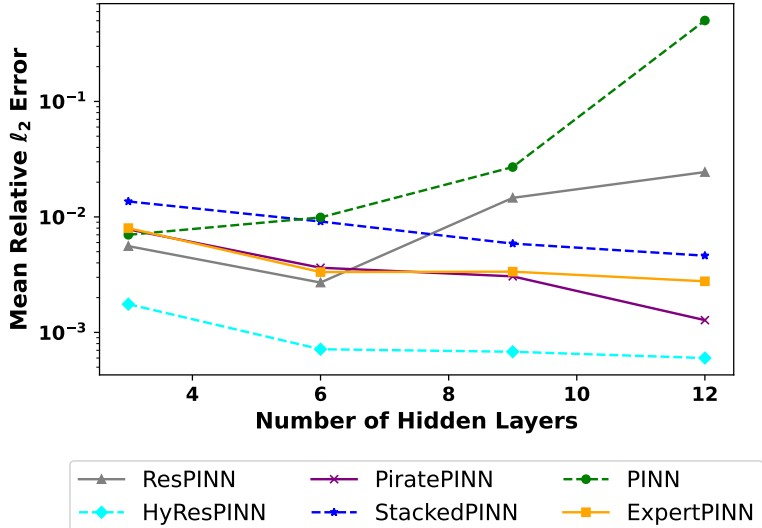

Figure 8: *Allen-Cahn equation (Periodic BCs):* Comparison of the mean relative $\ell_2$ error using various methods as a function of the number of hidden layers.

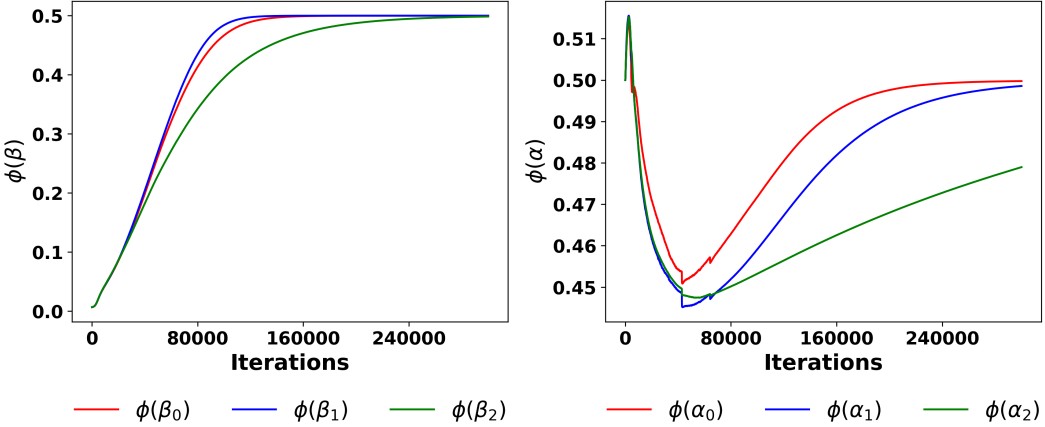

Figure 9: *Allen–Cahn equation (Periodic BCs):* Evolution of the learned gating parameters during training. (Left) Evolution of residual gating parameters $\phi(\beta_i)$, which modulate the strength of residual information flow across blocks. (Right) Evolution of hybrid gating parameters $\phi(\alpha_i)$, which balance the DNN and RBF components within each block.

consistently lower errors, outperforming the competing methods. Residual learning allows deeper networks to refine predictions iteratively, progressively improving both global structure and sharp transitions. The competing methods struggle to maintain accuracy as depth increases, whereas HyResPINNs remains robust, demonstrating the effectiveness of deep hybrid residual learning. Specifically, the standard PINNs prediction error increases as the network becomes deeper, eventually failing to capture any part of the solution. ResNets errors also increase with increased depth, though at a slower rate than the standard PINN. This is likely due to the static skip connection in the standard residual architecture design, preventing the adaptive learning of residual connections.

Figure 9 illustrates how the learned gating parameters evolve during training of HyResPINNs. The residual gating parameters $\phi(\beta)$ increase steadily during training, indicating the progressive strengthening of residual information flow across the blocks. The hybrid gating parameters $\phi(\alpha)$ initially decrease—reflecting a

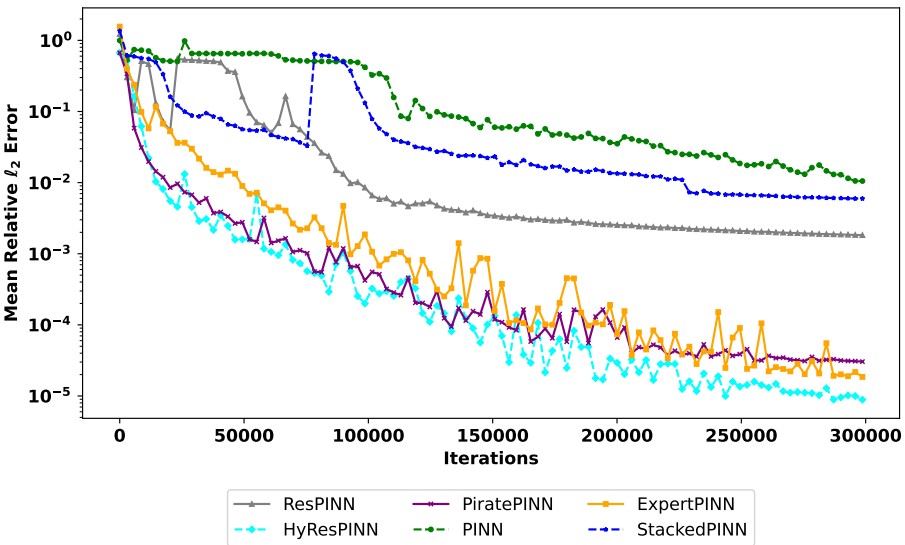

Figure 10: *Allen-Cahn equation (Periodic BCs):* Comparison of mean relative $\ell_2$ error across various methods, plotted against the number of training iterations.

stronger reliance on the RBFNN components—before stabilizing at values that indicate a moderate preference for the RBFNN.

*Efficiency: Accuracy and Convergence per Degree of Freedom.* Next, we assess the empirical efficiency of HyResPINNs relative to competing PINN baselines under comparable parameter budgets. As shown in Figure 10, HyResPINNs initially converge at a similar rate to PiratePINNs, but ultimately achieve a slightly lower mean relative $\ell_2$ error than all other baselines. Table 4 further summarizes the efficiency trade-offs defined in Equation 19 and Equation 20. While HyResPINNs exhibit the highest computational cost among the baselines ($T_{train} = 827$), they achieve the lowest mean relative $\ell_2$ error of $1.19e-5$. Consequently, HyResPINNs attain the highest parameter efficiency ($\eta_{\text{params}} = 1.106e-1$), and the highest computational efficiency ($\eta_{\text{comp}} = 1.016e2$), with the overall composite efficiency reaching $\eta_{\text{DoF}} = 1.337e-4$. Although $\eta_{\text{comp}}$ is only moderately higher than that of ExpertPINN ($\eta_{\text{comp}} = 1.979e1$), this improvement—combined with the lower error—demonstrates that HyResPINNs achieve superior accuracy-efficiency balance without requiring additional model capacity.

*Summary.* These results demonstrate that the hybrid, gated architecture of HyResPINNs enables improved accuracy and efficiency compared to conventional PINN variants. Despite comparable parameter counts, HyResPINNs achieve lower errors, underscoring the benefits of hybridization and the enhanced representational depth provided by the adaptive residual connections.

## 5.3 Darcy Flow with Smooth Coefficients

We next test the performance of HyResPINN on the Darcy Flow equation with smooth coefficients. Darcy Flow serves as a fundamental model for various physical processes, including porous media flow, heat transfer, and semiconductor doping. In most applications, the flux $\mathbf{u} = -\mu\nabla\phi$ is the variable of primary interest. The Darcy Flow problem is an elliptic boundary value problem given by:

$$-\nabla \cdot \mu\nabla\phi = f \quad \text{in } \Omega \tag{21}$$
$$\phi = u \quad \text{on } \Gamma_D$$
$$\mathbf{n} \cdot \mu\nabla\phi = g \quad \text{on } \Gamma_N$$

where $\Gamma_D$ and $\Gamma_N$ denote Dirichlet and Neumann parts of the boundary $\Gamma$, respectively, $\mu$ is a symmetric positive definite tensor describing a material property and $f$, $u$ and $g$ are given data.

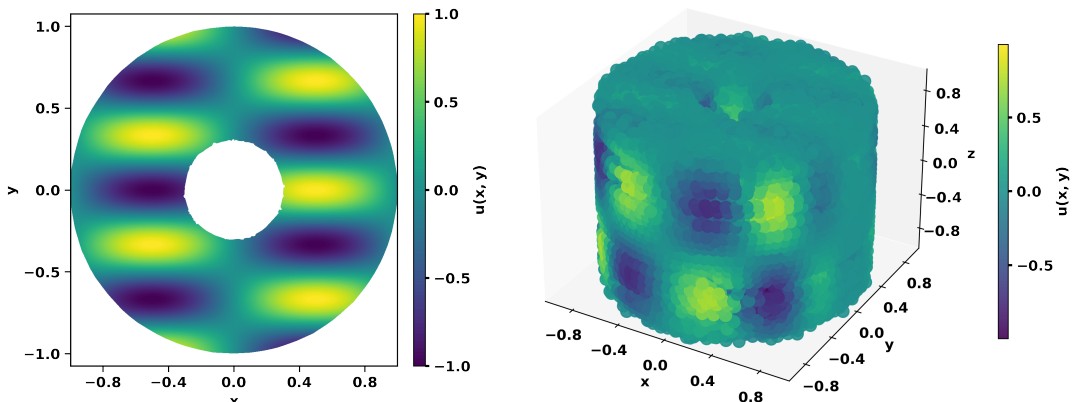

Figure 11: *Smooth Darcy Flow equation:* True solutions of the 2D and 3D problems.

We use manufactured solutions in both 2D and 3D domains, ensuring that the exact solution contains global structure and cross-dimensional correlations. Specifically, the exact solutions are given as:

$$u(x, y) = \sin(\pi x)\cos(3\pi y), \tag{22}$$

$$u(x, y, z) = \sin(\pi x)\sin(3\pi y)\sin(\pi z), \tag{23}$$

for the 2D and 3D cases, respectively. Figure 11 displays the target solutions for both cases.

To rigorously evaluate data efficiency, adaptivity, and convergence, we generated static training datasets for all experiments, fixing the set of collocation points within each run. This design enables controlled studies of error reduction as the number of training points increases, while eliminating the variability introduced by stochastic sampling. We systematically varied the number of collocation points to perform convergence studies, and solved each problem under both Dirichlet and Neumann boundary conditions in both 2D and 3D domains. This approach allows for a fair, direct comparison of accuracy, data efficiency, and robustness across problem dimensions and boundary types. All baseline models were matched in parameter count and trained under identical conditions for consistent benchmarking.

*Expressivity: Accurate Solution of Smooth Multidimensional PDEs.* HyResPINN accurately reconstructs these solutions and achieves low mean relative $\ell_2$ errors across all tested training set sizes (see Figure 12, solid lines), indicating its capacity to represent smooth solutions in higher dimensions. Notably, HyResPINN's predictions for both the solution and the physically relevant flux quantity remain stable and accurate even as dimensionality increases.

*Adaptivity: Robustness to Boundary Conditions and Domain Complexity.* HyResPINN maintains high accuracy across both types of boundary conditions in 2D and 3D, whereas baseline methods often exhibit significant degradation or instability, particularly for Neumann constraints (see Figure 12, where dashed lines denote flux errors). The model also scales from the 2D annulus to the more complex 3D cylindrical domain, demonstrating flexibility in handling increased geometric and dimensional complexity without additional tuning.

Figure 13 illustrates the evolution of the learned gating parameters for the 2D Dirichlet boundary condition case. The residual gating parameters $\phi(\beta)$ increase steadily during training, indicating the progressive strengthening of residual information flow across the blocks. The hybrid gating parameters $\phi(\alpha)$ initially decrease—reflecting a stronger reliance on the RBFNN components—before stabilizing at values that indicate a moderate preference for the RBFNN. Similar gating dynamics were observed for the 2D Neumann case and the 3D cases, as reported in Appendix C.

*Efficiency per Degree of Freedom: Data Efficiency and Convergence with Static Datasets.* Table 5 and Figure 12 show that HyResPINNs achieve lower or comparable solution and flux errors to competing models across nearly all configurations, except in the smallest-data regime, where ResPINNs attain the lowest mean

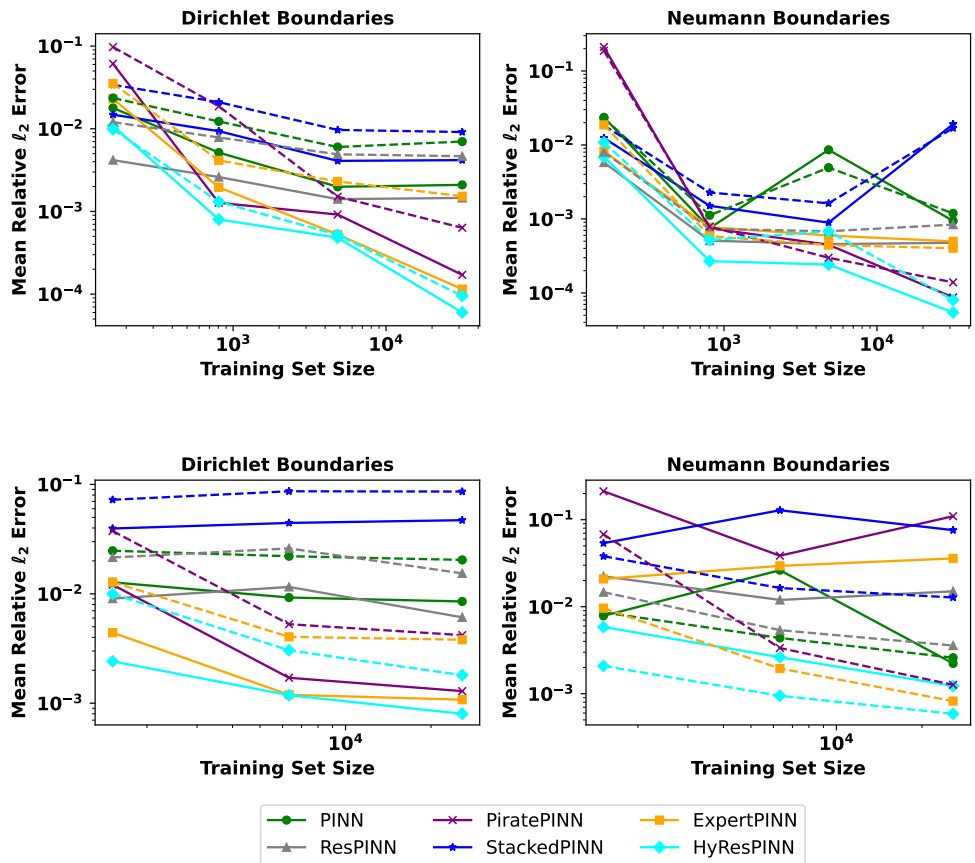

Figure 12: *2D and 3D Smooth Darcy Flow equations:* Comparison of the mean relative $\ell_2$ errors across each baseline method for the 2D Darcy Flow problem (top) and the 3D Darcy flow problem (bottom) plotted against the number of training collocation points using both Dirichlet and Neumann boundary conditions. Solid lines show the solution error, while dashed lines show the x-directional flux errors.

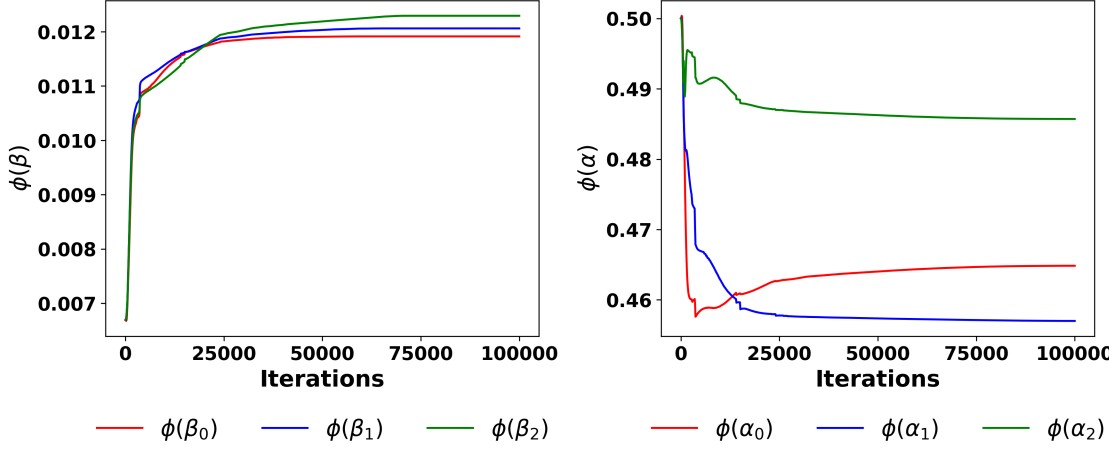

Figure 13: *2D Smooth Darcy-Flow equation (Dirichlet BCs):* Evolution of the learned gating parameters during training. (Left) Evolution of residual gating parameters $\phi(\beta_i)$, which modulate the strength of residual information flow across blocks. (Right) Evolution of hybrid gating parameters $\phi(\alpha_i)$, which balance the DNN and RBF components within each block. Similar convergence trends were observed for the 2D Neumann case and the 3D cases (see Appendix C).

| Model | $N_{params}$ | $T_{train}$ | Rel $\ell_2$ Error | $\eta_{params}$ | $\eta_{comp}$ | $\eta_{DoF}$ |
|---|---|---|---|---|---|---|
| **2D Smooth Darcy Flow** | | | | | | |
| **Dirichlet BCs** | | | | | | |
| PINN | 14,913 | 196 (7) | 7.54e-04 (0.00010) | 8.889e-2 | 6.760e0 | 4.533e-4 |
| ExpertPINN | 15,298 | 114 (5) | 5.09e-04 (0.00003) | 1.285e-1 | 1.730e1 | 1.131e-3 |
| PiratePINN | 15,302 | 115 (3) | 8.71e-05 (0.00001) | 7.503e-1 | 9.989e1 | 6.528e-3 |
| ResPINN | 12,737 | 196 (12) | 4.62e-04 (0.00009) | 1.699e-1 | 1.104e1 | 8.667e-4 |
| StackedPINN | 13,642 | 122 (3) | 9.00e-04 (0.00007) | 8.145e-2 | 9.111e0 | 6.679e-4 |
| HyResPINN | 14,248 | 125 (2) | 5.44e-05 (0.00002) | 1.290e0 | 1.471e2 | 1.032e-2 |
| **Neumann BCs** | | | | | | |
| PINN | 14,913 | 192 (6) | 2.05e-03 (0.00040) | 3.271e-2 | 2.537e0 | 1.701e-4 |
| ExpertPINN | 15,298 | 112 (5) | 1.29e-04 (0.00003) | 5.083e-1 | 6.921e1 | 4.524e-3 |
| PiratePINN | 15,302 | 112 (3) | 1.71e-04 (0.00003) | 3.822e-1 | 5.205e1 | 3.401e-3 |
| ResPINN | 12,737 | 192 (11) | 1.40e-03 (0.00007) | 5.600e-2 | 3.709e0 | 2.912e-4 |
| StackedPINN | 13,642 | 120 (2) | 4.10e-03 (0.00009) | 1.788e-2 | 2.024e0 | 1.484e-4 |
| HyResPINN | 14,248 | 123 (2) | 6.00e-05 (0.00004) | 1.170e0 | 1.350e2 | 9.475e-3 |
| **3D Smooth Darcy Flow** | | | | | | |
| **Dirichlet BCs** | | | | | | |
| PINN | 14,945 | 189 (6) | 2.05e-03 (0.00010) | 3.257e-2 | 2.580e0 | 1.726e-4 |
| ExpertPINN | 15,266 | 111 (4) | 2.10e-02 (0.00034) | 3.118e-3 | 4.284e-1 | 2.806e-5 |
| PiratePINN | 15,270 | 110 (3) | 3.90e-02 (0.00029) | 1.679e-3 | 2.333e-1 | 1.528e-5 |
| ResPINN | 12,753 | 189 (11) | 1.21e-02 (0.00076) | 6.501e-3 | 4.394e-1 | 3.445e-5 |
| StackedPINN | 13,834 | 119 (2) | 5.49e-02 (0.00040) | 1.317e-3 | 1.530e-1 | 1.106e-5 |
| HyResPINN | 14,344 | 115 (2) | 1.25e-03 (0.00002) | 5.575e-2 | 6.958e0 | 4.851e-4 |
| **Neumann BCs** | | | | | | |
| PINN | 14,945 | 185 (6) | 8.25e-03 (0.00023) | 8.106e-3 | 6.542e-1 | 4.377e-5 |
| ExpertPINN | 15,266 | 110 (4) | 1.00e-03 (0.00034) | 6.547e-2 | 9.095e0 | 5.957e-4 |
| PiratePINN | 15,270 | 109 (3) | 1.31e-03 (0.00019) | 5.010e-2 | 7.038e0 | 4.609e-4 |
| ResPINN | 12,753 | 185 (10) | 6.10e-03 (0.00071) | 1.285e-2 | 8.850e-1 | 6.939e-5 |
| StackedPINN | 13,834 | 119 (2) | 3.90e-02 (0.00023) | 1.853e-3 | 2.154e-1 | 1.557e-5 |
| HyResPINN | 14,344 | 115 (2) | 1.00e-03 (0.00003) | 6.969e-2 | 8.697e0 | 6.063e-4 |

Table 5: *2D and 3D Smooth Darcy Flow equations:* Comparison of model complexity and efficiency across PINN variants for both Dirichlet and Neumann boundary conditions for the largest-data regime. Reported are the total number of trainable parameters ($N_{params}$); the computational cost, expressed as the average number of training iterations completed per second ($T_{train}$) with standard deviation in parentheses; the mean relative $\ell_2$ error with standard deviation in parentheses; and the corresponding efficiency metrics $\eta_{params}$, $\eta_{comp}$, and $\eta_{DoF}$. Higher efficiency values indicate models that achieve lower relatives errors relative to their representational and computational costs.

relative $\ell_2$ errors for both 2D Dirichlet and Neumann cases. The advantages of HyResPINNs become most evident in the large-data regime—specifically, as the dataset size increases, HyResPINNs continue to reduce error, whereas other models plateau or even degrade in accuracy. When accounting for both parameter count and computational cost, HyResPINNs outperform the other baselines, achieving the highest composite efficiency metric ($\eta_{DoF}$) across all cases. In terms of computational efficiency ($\eta_{comp}$), HyResPINNs achieve the highest values in all but the 3D Neumann case, where ExpertPINNs perform slightly better. Nevertheless,

HyResPINNs consistently exhibit the largest $\eta_{\text{params}}$, indicating that they maintain balanced parameter efficiency across problem dimensions and boundary conditions.

*Summary.* HyResPINN demonstrates (i) strong function space expressivity for smooth multidimensional PDEs, (ii) robust adaptivity to varying boundary conditions and domain complexity, and (iii) superior efficiency per degree of freedom—achieving competitive accuracy and convergence across a range of training set sizes. These results further validate the hybrid residual framework as an effective and scalable solver for elliptic PDEs in both 2D and 3D settings.

## 5.4 Darcy Flow with Rough Coefficients

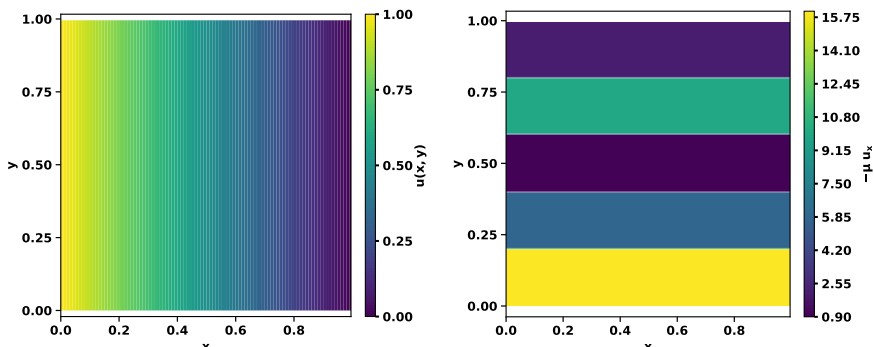

Figure 14: (Left) True solution on the 2D box domain. (Right) True transverse flux.

Accurately solving the Darcy Flow equation with discontinuous coefficients is a fundamental challenge for both numerical and machine learning-based PDE solvers. In heterogeneous materials, the solution and flux field must correctly capture interface behavior: while the normal component of the flux remains continuous, the tangential component may be discontinuous. We focus on the well-studied "five strip problem," well-documented manufactured solution test (Trask et al., 2017; Nakshatrala et al., 2006; Masud & Hughes, 2002), which evaluates the ability of a method to preserve the continuity of normal flux across material interfaces. This problem divides the domain $\Omega = [0,1]^2$ into five horizontal strips, each with a distinct diffusion coefficient $\mu_i$. Specifically, the prescribed exact solution on domain $\Omega = [0,1]^2$ under Neumann boundary conditions is given by:

$$\phi_{ex} = 1 - x, \quad \text{and} \quad \Gamma_N = \Gamma, \tag{24}$$

such that $\Omega$ is divided in five equal strips,

$$\Omega_i = \{(x,y) \mid 0.2(i-1) \leq y \leq 0.2i \, ; \, 0 \leq x \leq 1\}, \quad i = 1,...,5 \tag{25}$$

with different $\mu_i$ on each $\Omega_i$ such that $\mu_1 = 16$, $\mu_2 = 6$, $\mu_3 = 1$, $\mu_4 = 10$, $\mu_5 = 2$. Figure 14 shows the true solution and corresponding transverse flux.

*Expressivity: Capturing Discontinuous and Physically Consistent Flux.* The rough-coefficient Darcy Flow problem requires a model capable of representing changes in material properties and the resulting discontinuous flux behavior. HyResPINN's compositional block representation enables it to accurately approximate both the solution and its flux, even in the presence of abrupt material transitions.

*Adaptivity: Robustness to Material Interfaces.* Robust adaptivity is crucial for eliminating spurious oscillations and enforcing correct flux continuity at interfaces. Unlike classical collocated methods—which frequently exhibit oscillatory flux or fail to preserve normal continuity at discontinuities—HyResPINN adapts its representation to maintain the correct physical behavior of both normal and tangential flux components. The method handles Neumann boundary conditions and variable $\mu_i$, confirming its architectural flexibility for rough, heterogeneous domains.

Figure 15 illustrates the evolution of the learned gating parameters during training. The residual gating parameters, $\phi(\beta)$, increase steadily during training, indicating a progressive strengthening of residual information flow across network blocks. The hybrid gating parameters, $\phi(\alpha)$, initially decrease—reflecting an early reliance on the RBFNN components—before stabilizing to block-dependent values: blocks 0 and 2 converge to moderate preferences for the RBFNN, whereas block 1 stabilizes at a slightly higher $\phi(\alpha_1)$, indicating a mild preference for the DNN component.

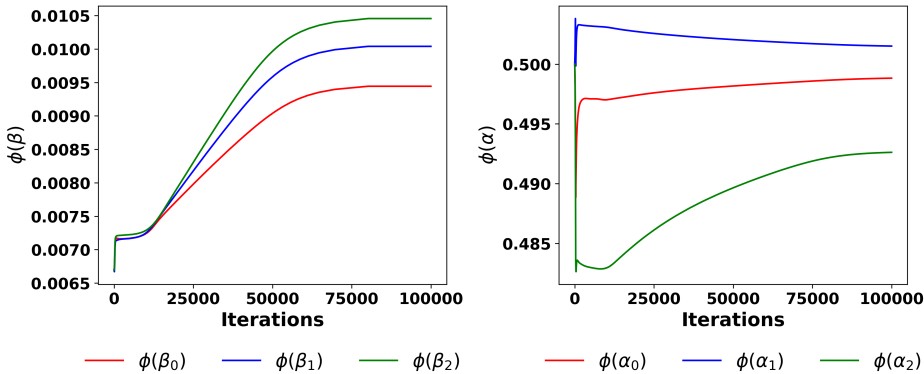

Figure 15: *2D Rough Darcy-Flow equation:* Evolution of the learned gating parameters during training. (Left) Evolution of residual gating parameters $\phi(\beta_i)$, which modulate the strength of residual information flow across blocks. (Right) Evolution of hybrid gating parameters $\phi(\alpha_i)$, which balance the DNN and RBF components within each block.

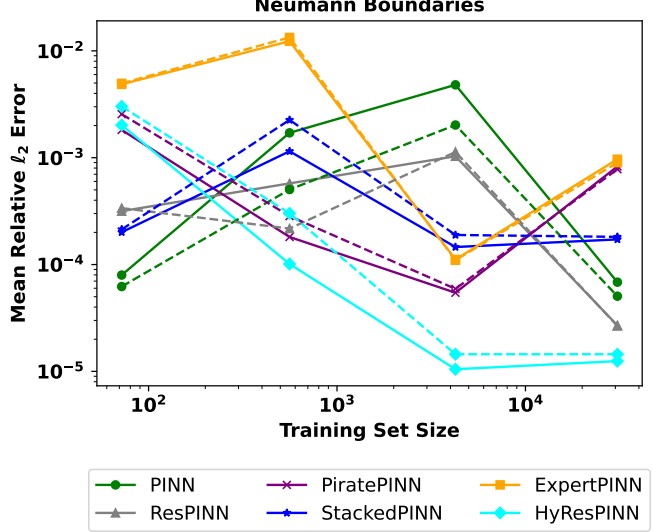

Figure 16: *2D Rough Darcy Flow equation:* Comparison of the mean relative $\ell_2$ errors across various methods, plotted against the number of training collocation points. Solid lines show the solution error, while dashed lines show the x-directional flux errors.

*Efficiency per Degree of Freedom: Convergence with Increasing Data.* To evaluate data efficiency, we perform convergence studies by progressively increasing the number of collocation points while maintaining comparable parameter counts across all models. Figure 16, HyResPINN achieves lower solution and flux errors as data size increases, similar to Section 5.3. While convergence on rough problems requires more data, HyResPINNs ultimately reach the lowest relative $\ell_2$ error. Furthermore, Table 6 reports the efficiency metrics across all

| Model | $N_{\text{params}}$ | $T_{\text{train}}$ | Rel $\ell_2$ Error | $\eta_{\text{params}}$ | $\eta_{\text{comp}}$ | $\eta_{\text{DoF}}$ |
|---|---|---|---|---|---|---|
| PINN | 14,913 | 196 (2) | 6.84e-05 (0.00001) | 9.803e-1 | 7.456e1 | 5.000e-3 |
| ExpertPINN | 15,298 | 114 (1) | 1.11e-04 (0.00003) | 5.910e-1 | 7.957e1 | 5.201e-3 |
| PiratePINN | 15,302 | 115 (1) | 5.44e-05 (0.00001) | 1.201e0 | 1.599e2 | 1.045e-2 |
| ResPINN | 12,737 | 196 (2) | 2.69e-05 (0.00001) | 2.919e0 | 1.896e2 | 1.489e-2 |
| StackedPINN | 13,642 | 122 (2) | 1.50e-04 (0.00001) | 4.887e-1 | 5.467e1 | 4.007e-3 |
| HyResPINN | 14,248 | 125 (1) | 1.05e-05 (0.00001) | 6.684e0 | 7.619e2 | 5.347e-2 |

Table 6: *2D Rough Darcy Flow equation:* Comparison of model complexity and efficiency across PINN variants for the largest-data regime. Reported are the total number of trainable parameters ($N_{\text{params}}$); the computational cost, expressed as the average number of training iterations completed per second ($T_{\text{train}}$) with standard deviation in parentheses; the mean relative $\ell_2$ error with standard deviation in parentheses; and the corresponding efficiency metrics $\eta_{\text{params}}$, $\eta_{\text{comp}}$, and $\eta_{\text{DoF}}$. Higher efficiency values indicate models that achieve lower relatives errors relative to their representational and computational costs.

baselines, showing that HyResPINNs achieve the highest overall efficiency ($\eta_{\text{DoF}} = 5.347e-2$), along with the largest computational efficiency ($\eta_{\text{comp}} = 7.619e2$), and parameter efficiency ($\eta_{\text{params}} = 6.684e0$).

*Summary.* HyResPINN demonstrates (i) strong expressivity for non-smooth PDEs, (ii) robust adaptivity to discontinuous material interfaces, and (iii) high efficiency per DoF—achieving physically consistent solutions and fluxes in rough-coefficient Darcy Flow problems.

### 5.5 Kuramoto–Sivashinsky Equation

In this section, we use the Kuramoto–Sivashinsky equation as a benchmark problem under periodic boundary conditions. The Kuramoto–Sivashinsky equation is a fourth-order nonlinear partial differential equation widely used to model spatiotemporal patterns in fluid dynamics, combustion, and other systems exhibiting chaotic behavior. We consider the 1D Kuramoto–Sivashinsky equation of the form:

$$\frac{\partial u}{\partial t} + \alpha u \frac{\partial u}{\partial x} + \beta \frac{\partial^2 u}{\partial x^2} + \gamma \frac{\partial^4 u}{\partial x^4} = 0, \tag{26}$$

subject to periodic boundary conditions and an initial condition:

$$u(0, x) = u_0(x) = \cos(x)(1 + \sin(x)). \tag{27}$$

The parameters $\alpha$, $\beta$, and $\gamma$ control the dynamical behavior of the equation. We use the configurations: $\alpha = 100/16$, $\beta = 100/16^2$, and $\gamma = 100/16^4$ for chaotic behaviors. Solving this equation allows us to assess HyResPINN's ability to approximate highly complex and chaotic systems. Following (Wang et al., 2023a), we employ a time-marching training routine to facilitate stable optimization. HyResPINN and all baseline models are trained using the same time-stepping scheme and comparable parameter budgets. Full experimental details are provided in Appendix D.

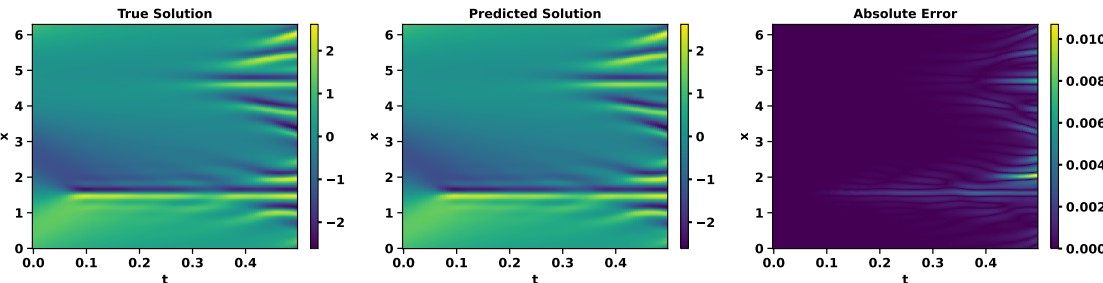

Figure 17: *Kuramoto–Sivashinsky equation:* Comparison of the best prediction against the reference solution.

| Model | $\mathbf{N}_{\text{params}}$ | $\mathbf{T}_{\text{train}}$ | Rel $\ell_2$ Error | $\boldsymbol{\eta}_{\text{params}}$ | $\boldsymbol{\eta}_{\text{comp}}$ | $\boldsymbol{\eta}_{\text{DoF}}$ |
|---|---|---|---|---|---|---|
| PINN | 724,353 | 1961 (7) | 1.42e-01 (0.00300) | 9.717e-6 | 3.590e-3 | 4.956e-9 |
| ExpertPINN | 727,170 | 1553 (9) | 1.57e-03 (0.00024) | 8.756e-4 | 4.099e-1 | 5.637e-7 |
| PiratePINN | 727,171 | 1520 (5) | 2.00e-03 (0.00020) | 6.876e-4 | 3.289e-1 | 4.523e-7 |
| ResPINN | 790,145 | 1631 (8) | 1.56e-02 (0.00090) | 8.113e-5 | 3.929e-2 | 4.973e-8 |
| StackedPINN | 793,738 | 909 (1) | 1.86e-01 (0.00210) | 6.773e-6 | 5.914e-3 | 7.451e-9 |
| HyResPINN | 759,686 | 909 (1) | 8.12e-04 (0.00072) | 1.620e-3 | 1.354e0 | 1.782e-6 |

Table 7: *Kuramoto–Sivashinsky equation:* Comparison of model complexity and efficiency across PINN variants. Reported are the total number of trainable parameters ($N_{\text{params}}$); the computational cost, expressed as the average number of training iterations completed per second ($T_{\text{train}}$) with standard deviation in parentheses; the mean relative $\ell_2$ error with standard deviation in parentheses; and the corresponding efficiency metrics $\eta_{\text{params}}$, $\eta_{\text{comp}}$, and $\eta_{\text{DoF}}$. Higher efficiency values indicate models that achieve lower relatives errors relative to their representational and computational costs.

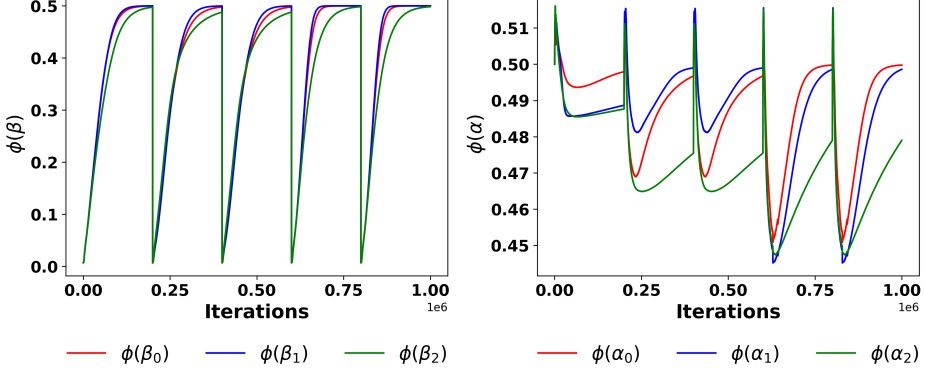

Figure 18: *Kuramoto–Sivashinsky equation:* Evolution of the learned gating parameters during training. (Left) Evolution of residual gating parameters $\phi(\beta_i)$, which modulate the strength of residual information flow across blocks. (Right) Evolution of hybrid gating parameters $\phi(\alpha_i)$, which balance the DNN and RBF components within each block.

*Expressivity: Capturing Chaotic and Multi-scale Dynamics.* The Kuramoto-Sivashinsky equation presents chaotic, multi-scale temporal behaviour and sharp spatial gradients that challenge standard PINN architectures. HyResPINN demonstrates strong expressivity by accurately recovering the chaotic solutions generated by the Kuramoto–Sivashinsky equation, closely matching the reference solution shown in Figure 17.

*Adaptivity: Robustness to evolving spatiotemporal features.* HyResPINNs adapt to evolving chaotic spatial features, maintaining a high accuracy relative to the baselines through the time-marching scheme. Figure 18 illustrates the evolution of the learned gating parameters under the time-marching training routine. The residual gating parameters $\phi(\beta)$ increase rapidly within each time segment and then reset at the start of the next window, reflecting the progressive strengthening and reinitialization of residual information flow across blocks. In contrast, the hybrid gating parameters $\phi(\alpha)$ initially decrease and then stabilize within each window, indicating a dominance of the RBFNN components before rebalancing toward a mixed DNN–RBFNN representation. These cyclic patterns in $\phi(\alpha)$ and $\phi(\beta)$ highlight how HyResPINNs dynamically adapt their representational balance at each stage of temporal evolution, enabling stable and accurate long-term integration of chaotic PDEs.

*Efficiency per Degree of Freedom.* In terms of efficiency per DoF, HyResPINNs achieve the lowest mean relative $\ell_2$ error ($8.12e{-}4$) among all tested methods, as summarized in Table 7. They also obtain the highest parameter efficiency ($\eta_{\text{params}} = 1.620e-3$), computational efficiency ($\eta_{\text{comp}} = 1.354e0$), and composite efficiency ($\eta_{\text{DoF}} = 1.782e{-}6$), indicating strong accuracy relative to representational complexity and training cost. These results highlight that HyResPINNs sustain high predictive accuracy even in chaotic regimes where other PINN architectures deteriorate.

*Summary.* HyResPINNs demonstrate (i) strong expressivity for chaotic, spatiotemporally complex PDEs, (ii) robust adaptivity to evolving multi-scale features, and (iii) high empirical efficiency per DoF—achieving accurate, physically consistent predictions for the Kuramoto-Sivashinsky equation.

## 5.6 Korteweg-De Vries Equation

Next, we explore the one-dimensional Korteweg–De Vries (KdV) equation, a fundamental model used to describe the dynamics of solitary waves, or solitons. The KdV equation is expressed as follows:

$$u_t + \eta u u_x + \mu^2 u_{xxx} = 0, \quad t \in (0,1), \quad x \in (-1,1),$$
$$u(x,0) = \cos(\pi x),$$
$$u(t,-1) = u(t,1),$$

where $\eta$ governs the strength of the nonlinearity and $\mu$ controls the dispersion level. Under the KdV dynamics, this initial wave evolves into a series of solitary-type waves. We adopt the classical parameters of the KdV equation, setting $\eta = 1$ and $\mu = 0.022$ (Zabusky & Kruskal, 1965). Full experimental details are provided in Appendix D.

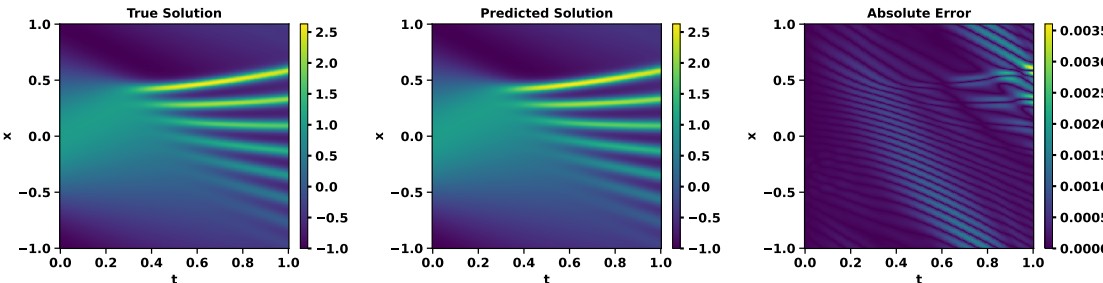

Figure 19: *Korteweg-De Vries equation:* Comparison of the best prediction against the reference solution.

| Model | $N_{\text{params}}$ | $T_{\text{train}}$ | Rel $\ell_2$ Error | $\eta_{\text{params}}$ | $\eta_{\text{comp}}$ | $\eta_{\text{DoF}}$ |
|---|---|---|---|---|---|---|
| PINN | 724,353 | 1947 (6) | 8.17e-01 (0.00300) | 1.689e-6 | 6.285e-4 | 8.676e-10 |
| ExpertPINN | 727,170 | 1553 (9) | 6.87e-04 (0.00010) | 2.003e-3 | 9.377e-1 | 1.289e-6 |
| PiratePINN | 727,171 | 1520 (5) | 5.61e-04 (0.00004) | 2.451e-3 | 1.173e0 | 1.612e-6 |
| ResPINN | 790,145 | 1631 (8) | 5.73e-01 (0.00190) | 2.210e-6 | 1.071e-3 | 1.355e-9 |
| StackedPINN | 793,738 | 909 (1) | 2.98e-01 (0.00210) | 4.228e-6 | 3.693e-3 | 4.652e-9 |
| HyResPINN | 759,686 | 885 (1) | 4.09e-04 (0.00007) | 3.214e-3 | 2.759e0 | 3.632e-6 |

Table 8: *Korteweg-De Vries equation:* Comparison of model complexity and efficiency across PINN variants. Reported are the total number of trainable parameters ($N_{\text{params}}$); the computational cost, expressed as the average number of training iterations completed per second ($T_{\text{train}}$) with standard deviation in parentheses; the mean relative $\ell_2$ error with standard deviation in parentheses; and the corresponding efficiency metrics $\eta_{\text{params}}$, $\eta_{\text{comp}}$, and $\eta_{\text{DoF}}$. Higher efficiency values indicate models that achieve lower relatives errors relative to their representational and computational costs.

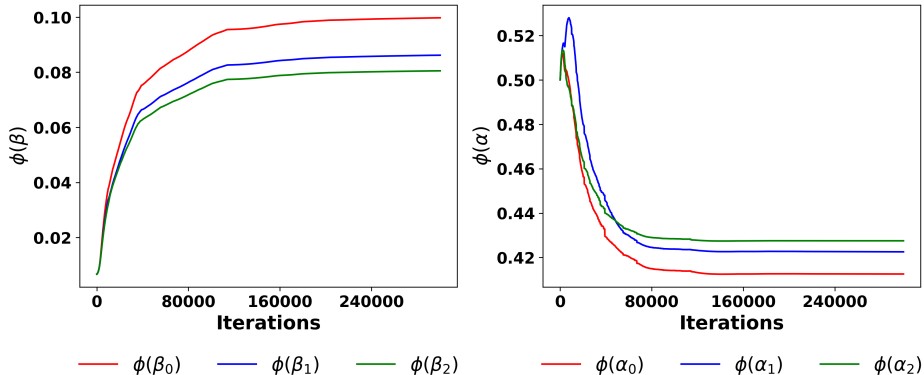

Figure 20: *Korteweg-De Vries equation:* Evolution of the learned gating parameters during training. (Left) Evolution of residual gating parameters $\phi(\beta_i)$, which modulate the strength of residual information flow across blocks. (Right) Evolution of hybrid gating parameters $\phi(\alpha_i)$, which balance the DNN and RBF components within each block.

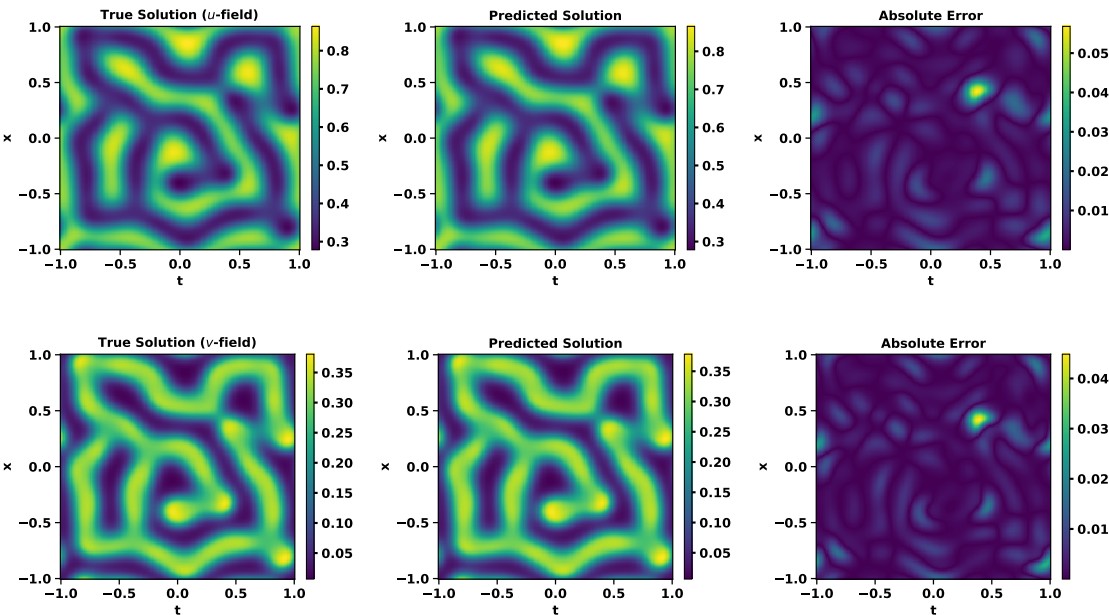

Figure 21: *Grey-Scott equation:* Comparisons between the solutions predicted by a trained HyResPINN and the reference solutions.

*Expressivity: Capturing Dispersive and Nonlinear Wave Dynamics.* The Korteweg-De Vries features strongly nonlinear, dispersive behaviour that produces stable solitary waves. HyResPINNs accurately reconstruct these soliton dynamics, as illustrated in Figure 19.

*Adaptivity: Preservation of Localized Wave Features.* The model adaptively resolves the highly localized soliton structures and evolving nonlinear features across the domain, demonstrating its ability to allocate expressive capacity where needed as the solution evolves. Figure 20 illustrates the evolution of the learned gating parameters. The residual gating parameters $\phi(\beta)$ increase steadily during training, indicating the progressive strengthening of residual information flow across the blocks. The hybrid gating parameters $\phi(\alpha)$ steadily decrease then stabilize—reflecting a stronger reliance on the RBFNN components.

*Efficiency per Degree of Freedom: Accuracy-Efficiency Trade-offs.* As summarized in Table 8, HyResPINNs achieve the lowest relative $\ell_2$ error ($4.09e - 4$) among all tested models. They also exhibit the highest empirical efficiency per degree of freedom, outperforming baselines with closely matched parameter budgets. These results confirm that HyResPINNs achieve high accuracy and efficient approximation of dispersive nonlinear PDEs, effectively balancing accuracy and computational cost.

*Summary.* HyResPINNs demonstrate strong expressivity for nonlinear, dispersive wave dynamics, robust adaptivity in preserving localized soliton structures, and high efficiency per DoF—achieving accurate and computationally efficient solutions for the Korteweg-De Vries equation.

## 5.7 Grey-Scott equation

In this example, we solve the 2D Grey-Scott equation, a reaction-diffusion system that describes the interaction of two chemical species. The form of this PDE is given as follows

$$u_t = \epsilon_1 \Delta u + b_1(1 - u) - c_1 u v^2, \quad t \in (0, 2), \ (x, y) \in (-1, 1)^2,$$
$$v_t = \epsilon_2 \Delta v - b_2 v + c_2 u v^2, \quad t \in (0, 2), \ (x, y) \in (-1, 1)^2,$$

| Model | $N_\text{params}$ | $T_\text{train}$ | Rel $\ell_2$ Error | $\eta_\text{params}$ | $\eta_\text{comp}$ | $\eta_\text{DoF}$ |
|---|---|---|---|---|---|---|
| **Grey–Scott: $u$-field solution** | | | | | | |
| PINN | 724,353 | 990 (17) | 7.54e-01 (0.00300) | 1.830e-6 | 1.339e-3 | 1.849e-9 |
| ExpertPINN | 727,170 | 804 (23) | 4.80e-03 (0.00030) | 2.867e-4 | 2.593e-1 | 3.566e-7 |
| PiratePINN | 727,171 | 795 (15) | 3.61e-03 (0.00040) | 3.809e-4 | 3.484e-1 | 4.791e-7 |
| ResPINN | 790,145 | 824 (20) | 9.80e-03 (0.00090) | 1.291e-4 | 1.238e-1 | 1.566e-7 |
| StackedPINN | 793,994 | 455 (4) | 1.21e-03 (0.00098) | 1.041e-3 | 1.818e0 | 2.290e-6 |
| HyResPINN | 759,814 | 455 (3) | 6.43e-03 (0.00090) | 2.047e-4 | 3.421e-1 | 4.503e-7 |
| **Grey–Scott: $v$-field solution** | | | | | | |
| PINN | 724,353 | 990 (17) | 9.56e-01 (0.00300) | 1.444e-6 | 1.056e-3 | 1.458e-9 |
| ExpertPINN | 727,170 | 804 (23) | 8.99e-03 (0.00030) | 1.530e-4 | 1.384e-1 | 1.903e-7 |
| PiratePINN | 727,171 | 795 (15) | 9.81e-03 (0.00040) | 1.402e-4 | 1.282e-1 | 1.763e-7 |
| ResPINN | 790,145 | 824 (20) | 1.10e-02 (0.00090) | 1.151e-4 | 1.103e-1 | 1.396e-7 |
| StackedPINN | 793,994 | 455 (4) | 6.90e-02 (0.00030) | 1.825e-5 | 3.188e-2 | 4.015e-8 |
| HyResPINN | 759,814 | 455 (3) | 1.31e-02 (0.00090) | 1.002e-4 | 1.676e-1 | 2.205e-7 |

Table 9: *Grey Scott equation:* Comparison of model complexity and efficiency across PINN variants for the $u$-field (top) and $v$-field (bottom) solutions. Reported are the total number of trainable parameters ($N_\text{params}$); the computational cost, expressed as the average number of training iterations completed per second ($T_\text{train}$) with standard deviation in parentheses; the mean relative $\ell_2$ error with standard deviation in parentheses; and the corresponding efficiency metrics $\eta_\text{params}$, $\eta_\text{comp}$, and $\eta_\text{DoF}$. Higher efficiency values indicate models that achieve lower relatives errors relative to their representational and computational costs.

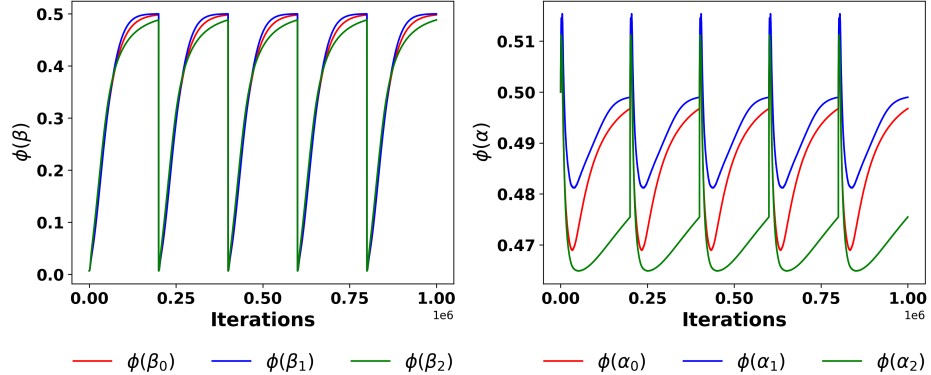

Figure 22: *Grey-Scott equation:* Evolution of the learned gating parameters during training. (Left) Evolution of residual gating parameters $\phi(\beta_i)$, which modulate the strength of residual information flow across blocks. (Right) Evolution of hybrid gating parameters $\phi(\alpha_i)$, which balance the DNN and RBF components within each block.

subject to the periodic boundary conditions and the initial conditions

$$u_0(x, y) = 1 - \exp(-10((x + 0.05)^2 + (y + 0.02)^2)),$$
$$v_0(x, y) = 1 - \exp(-10((x - 0.05)^2 + (y - 0.02)^2)).$$

Here $u$ and $v$ represent the concentrations of the two species. The system can generate a wide range of patterns, including spots, stripes, and more complex forms, depending on the parameters chosen. For this example, we set $\epsilon_1 = 0.2, \epsilon_2 = 0.1, b_1 = 40, b_2 = 100, c_1 = c_2 = 1,000$. We employ a standard time-marching strategy to train our PINN models, as outlined in various studies (Wight & Zhao, 2020; Wang et al., 2023a). Full experimental details are provided in Appendix F.

*Expressivity: Capturing Multi-component Patterns.* For the Grey-Scott reaction–diffusion system, HyResPINNs capture the essential spatiotemporal patterns. While not achieving the lowest overall errors among the tested models, the predicted $u$- and $v$-field solutions remain consistent with the reference patterns, as shown in Figure 21, demonstrating that the hybrid formulation can represent both localized and diffuse structures.

*Adaptivity: Evolving Spatiotemporal Features.* The complex, time-varying nature of the Grey–Scott system requires adaptive modeling of spatially localized features and their dynamic interactions. HyResPINNs dynamically resolve these interactions by adaptively allocating expressive capacity where needed as the solution evolves in each time-marching window. Figure 22 illustrates the evolution of the learned gating parameters under the time-marching training routine. The residual gating parameters $\phi(\beta)$ increase rapidly within each time segment and then reset at the start of the next window, reflecting the progressive strengthening and reinitialization of residual information flow across blocks. In contrast, the hybrid gating parameters $\phi(\alpha)$ initially decrease and then stabilize within each window, indicating a dominance of the RBFNN components before rebalancing toward a mixed DNN–RBFNN representation.

*Efficiency per Degree of Freedom: Accuracy-Efficiency Trade-offs.* As summarized in Table 9, HyResPINNs achieves competitive error rates compared to the baseline PINN models, with StackedPINNs attaining the lowest $u$-field relative $\ell_2$ error $(1.21e-3)$ and ExpertPINN the lowest $v$-field relative $\ell_2$ error of $(8.99e-3)$. While these results indicate that HyResPINNs are not the most accurate or computationally efficient model for this particular system, they remain comparable across all efficiency metrics. The reduced performance likely arises from the strong stiffness and multiscale coupling between the reaction and diffusion terms, which pose challenges for the hybrid RBFNN-DNN representation. In particular, the current gating structures may be too restrictive, limiting the model's ability to simultaneously capture diffusive behavior and rapidly varying reaction dynamics, making it difficult to maintain an optimal balance across space and time. Future work could address these limitations by introducing higher-dimensional adaptive skip connections, allowing the model to better accommodate the distinct scales present in stiff, multi-component systems.

*Summary.* HyResPINNs exhibit comparable accuracy for capturing the multi-component pattern formation present in the Grey-Scott problem. Although not the most accurate or computationally efficient model for this problem, the hybrid architecture remains competitive compared to the baseline methods. These results highlight the challenges posed by stiff, multiscale coupling in such systems and suggest that further adaptivity in the gating or skip-connection design could enhance performance in coupled problems.

## 6 Conclusion

In this work, we introduced HyResPINNs, a novel class of physics-informed neural networks that integrate adaptive hybrid residual blocks, combining the complementary strengths of DNNs and RBFNNs. The use of adaptive gating parameters allows the model to dynamically balance the contributions of DNN and RBFNN components throughout training, resulting in enhanced expressivity and adaptivity. Through our experiments on a diverse set of benchmark PDEs, we demonstrated that HyResPINNs generally outperforms state-of-the-art neural PDE solvers in accuracy and greater empirical efficiency, as quantified by our proposed metrics that account for error, computational cost, and representational capacity. Our method excels particularly in problems featuring mixed smooth and non-smooth solution features, where existing approaches often struggle.

**Limitations and Future Work.** While HyResPINNs demonstrate clear advantages in handling PDEs with sharp gradients and non-standard domains, several limitations remain. First, the inclusion of both RBF-based and DNN components increases the number of hyperparameters and architectural choices, introducing additional challenges in model selection and training. Hyperparameter tuning, in particular, can become computationally demanding for large-scale problems. Second, our current experiments are restricted to moderately sized PDEs; scaling HyResPINNs to higher-dimensional systems will require further investigation into both efficiency and numerical stability. Finally, extending the HyResPINN framework toward more general learning paradigms—such as operator learning—represents a promising direction for future work.

**Broader Impacts** This work contributes to the field of scientific machine learning by proposing a hybrid architecture for solving PDEs. The potential impact of HyResPINNs lies primarily in improving the accuracy of PDE solvers, which may accelerate progress in fields such as physics and engineering. As the method focuses on numerical modeling and does not directly process personal or sensitive data, we do not foresee any negative societal or ethical consequences.

### Acknowledgments

This research was supported by the U.S. Army Research Office (ARO) Award No. W911NF-22-2-0158. VS received support from the National Science Foundation under Award No. NSF SHF-2403379. We additionally acknowledge the computational resources provided by the University of Utah Center for High Performance Computing (CHPC) and the use of large language models (LLMs) for manuscript editing.

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

# A    Allen-Cahn with Dirichlet Boundary Conditions

## A.1    Reference Solution Generation

We solve the equation numerically using a semi-implicit finite difference method. The spatial domain is discretized uniformly with 1042 grid points, and the temporal domain is discretized into 600 steps. The nonlinear reaction term is treated explicitly, while the linear diffusion term is handled implicitly, forming an IMEX (implicit-explicit) Euler scheme. Homogeneous Dirichlet boundary conditions are imposed at each time step by modifying the linear system accordingly. Solutions are recorded by saving the solution $u(x,t)$ at all time steps along with the initial condition and spatial-temporal grids.

## A.2    Ablation Study: Impact of RBF Kernels and Center Density

To evaluate the impact of different radial basis function (RBF) kernels on solution quality, we conduct an ablation study comparing several anisotropic RBF formulations within the hybrid PINN framework. The goal is to determine how the kernel's support, smoothness, and decay behavior affect the model's ability to capture localized features, represent sharp interfaces, and maintain global solution coherence. The architecture, initialization, and training procedure are held constant across trials to isolate the influence of the kernel's functional form. Table 1 summarizes the five kernels studied.

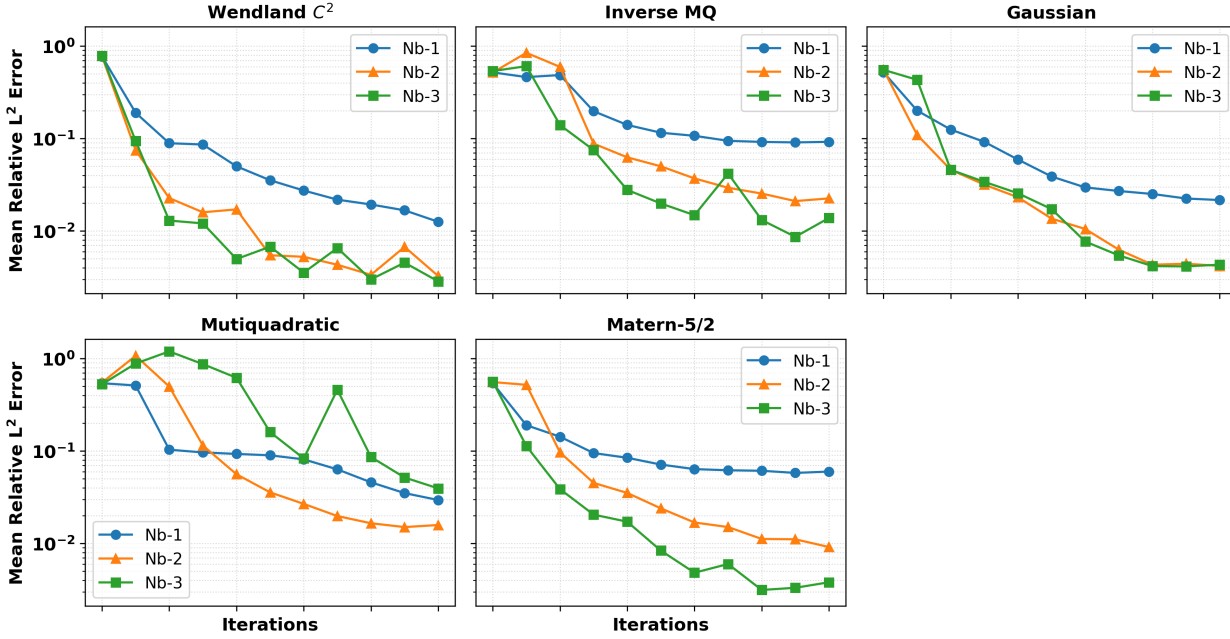

Figure 23: *Allen–Cahn equation:* Prediction errors for various RBF kernels within the HyResPINN framework using 1, 2, and 3 blocks.

Results demonstrate that kernel choice substantially influences the model's capacity to resolve steep gradients and preserve global coherence. The best overall performance is achieved by the Wendland $C^2$ kernel, which we adopt for all subsequent experiments, as it offers a slightly better trade-off between accuracy, computational efficiency, compact support, and $C^2$ smoothness which make it well-suited for the localized dynamics. In addition to kernel type, we also investigate how the number of RBF centers affects predictive performance, using the Wendland $C^2$ kernel. Results are shown in Table 11 which show that a balance between resolution and overfitting emerges, with best performance at 64 centers.

| Name | Computational Cost |
|---|---|
| Gaussian(Nb-1) | 0.020064 (0.00067) |
| Gaussian(Nb-2) | 0.040506 (0.00044) |
| Gaussian(Nb-3) | 0.062309 (0.00045) |
| Inverse MQ(Nb-1) | 0.020070 (0.00057) |
| Inverse MQ(Nb-2) | 0.040792 (0.00024) |
| Inverse MQ(Nb-3) | 0.062468 (0.00075) |
| Matérn-5/2(Nb-1) | 0.020033 (0.00076) |
| Matérn-5/2(Nb-2) | 0.040725 (0.00087) |
| Matérn-5/2(Nb-3) | 0.064149 (0.00043) |
| Multiquadric(Nb-1) | 0.019847 (0.00080) |
| Multiquadric(Nb-2) | 0.040620 (0.00077) |
| Multiquadric(Nb-3) | 0.062131 (0.00044) |
| Wendland $C^2$(Nb-1) | 0.021837 (0.00071) |
| Wendland $C^2$(Nb-2) | 0.043131 (0.00043) |
| Wendland $C^2$(Nb-3) | 0.065628 (0.00097) |

Table 10: Computational cost measured as the average time in seconds to run 100 iterations for each RBF using 1,2, and 3 blocks. Standard deviations are in parenthesis.

| Number of Centers | Error (Std) | Computational Cost |
|---|---|---|
| 8 | 0.028290 (0.042173) | 0.017529 (2.634898) |
| 16 | 0.007372 (0.002045) | 0.017578 (3.782532) |
| 32 | 0.003666 (0.001104) | 0.017849 (6.860452) |
| 64 | 0.002462 (0.000063) | 0.018479 (2.414555) |
| 128 | 0.003265 (0.001240) | 0.019347 (7.184156) |
| 256 | 0.002938 (0.000411) | 0.021444 (6.183969) |

Table 11: Effect of increasing the number of RBF centers on HyResPINN mean relative $\ell_2$ error and computational cost measured as the average wall-clock time (in seconds) required to perform 100 iterations. Standard deviations in parentheses.

## A.3 Ablation Study: Impact of Gating Parameter Initializations

To better understand the role of the learnable gating parameters in modulating the hybrid outputs and the residual skip connections, we conducted an ablation study varying the initial values of both gates ($\alpha_{\text{init}}$ and $\beta_{\text{init}}$). We tested the Allen–Cahn equation with Dirichlet boundary conditions from Section 5.1 as the benchmark due to its mixed smooth and sharp features. All other hyperparameters were kept fixed across runs, including the number of blocks ($N_b = 3$), hidden dimension ($d_h = 64$), and batch size (1024). Unless otherwise noted, we set the regularization strengths $\lambda_\alpha = \lambda_\beta = 1 \times 10^{-4}$ and training was performed for 200,000 epochs over five random seeds.

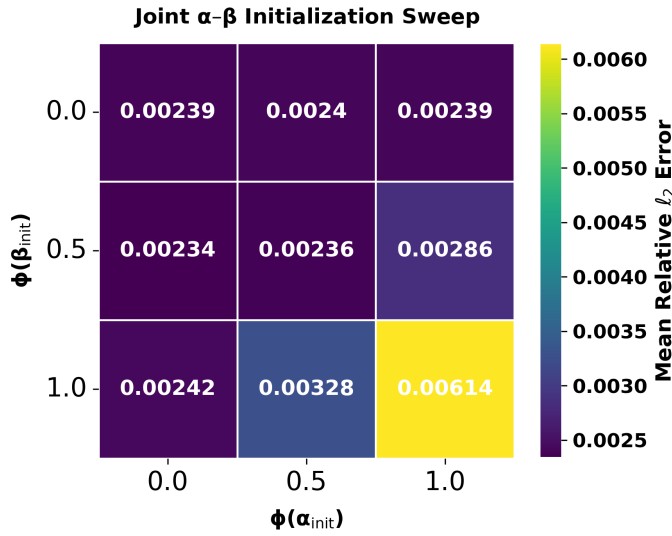

Figure 24: *Allen-Cahn equation:* Mean relative $\ell_2$ error for different combinations of $\phi(\alpha_{\text{init}})$ and $\phi(\beta_{\text{init}})$.

Figure 24 shows the joint effect of the gating initializations by reporting the mean relative $\ell_2$ errors across all $(\phi(\alpha_{\text{init}}), \phi(\beta_{\text{init}}))$ pairs in $\{0, 0.5, 1\}$. We observe a clear coupling between the two gating mechanisms: performance deteriorates when both gates are initialized near unity, suggesting excessive weighting of the DNN and transform pathways hinder learning. In contrast, intermediate or asymmetric initialization—such as $\phi(\alpha_{\text{init}}) \approx 0.5$ and $\phi(\beta_{\text{init}}) \approx 0$—yield the lowest test errors, suggesting that balanced initialization between block components (RBFNN and DNN, controlled by $\phi(\alpha)$) combined with an initially identity-favoring $\phi(\beta)$, enables the network to form a more effective mixture between the block components and to progressively learn deeper transforms through $\phi(\beta)$ during training.

To further isolate the individual effects of each gating mechanism, we visualize, in Figures 25 and 26, the final learned values of $\phi(\alpha)$ and $\phi(\beta)$ across block depth (0–2) for varying initializations, sigmoid temperatures ($t$), and regularization strengths ($\lambda_\alpha$, $\lambda_\beta$). The left-most subplots correspond to unconstrained cases where no sigmoid constraint ($\phi(\cdot)$) is applied meaning that the gating parameters are not constrained to lie in [0, 1], while the remaining columns apply progressively higher temperatures controlling the sharpness of the sigmoid function (see Section 4.1 for further details). In the top rows, line colors represent the initial parameter values; in the bottom rows, the same experiments are color-mapped by the resulting mean relative $\ell_2$ error.

Higher initialization or lower temperature leads to sharper, DNN-dominant gating ($\phi(\alpha) \to 1$), emphasizing DNN components, while lower initialization or stronger regularization biases the network toward RBFNN-dominant gating ($\phi(\alpha) \to 0$). Variations in $\phi(\alpha)$ across depth demonstrate the adaptive blending of local and global representations. In contrast, the $\beta$-gates primarily modulate residual flow: smaller $\phi(\beta)$ values promote identity mapping (residual-dominant) and larger values encourage deeper, transform-dominant behavior. Together, these results confirm that the gating parameters not only balance the RBFNN and DNN

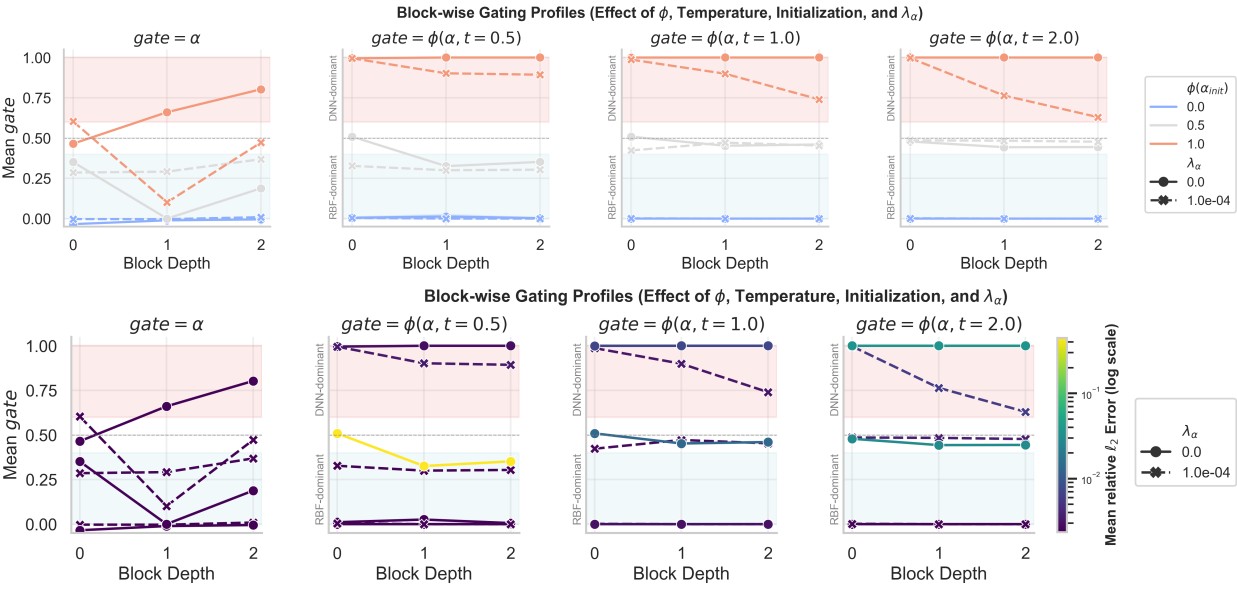

Figure 25: *Allen-Cahn equation:* Depth-wise $\alpha$ gating parameters for varying initialization, temperature, and regularization. $\alpha$-gates balance RBFNN and DNN representations, where larger gate values bias the output towards the DNN and smaller values to the RBFNN. The top figure shows the

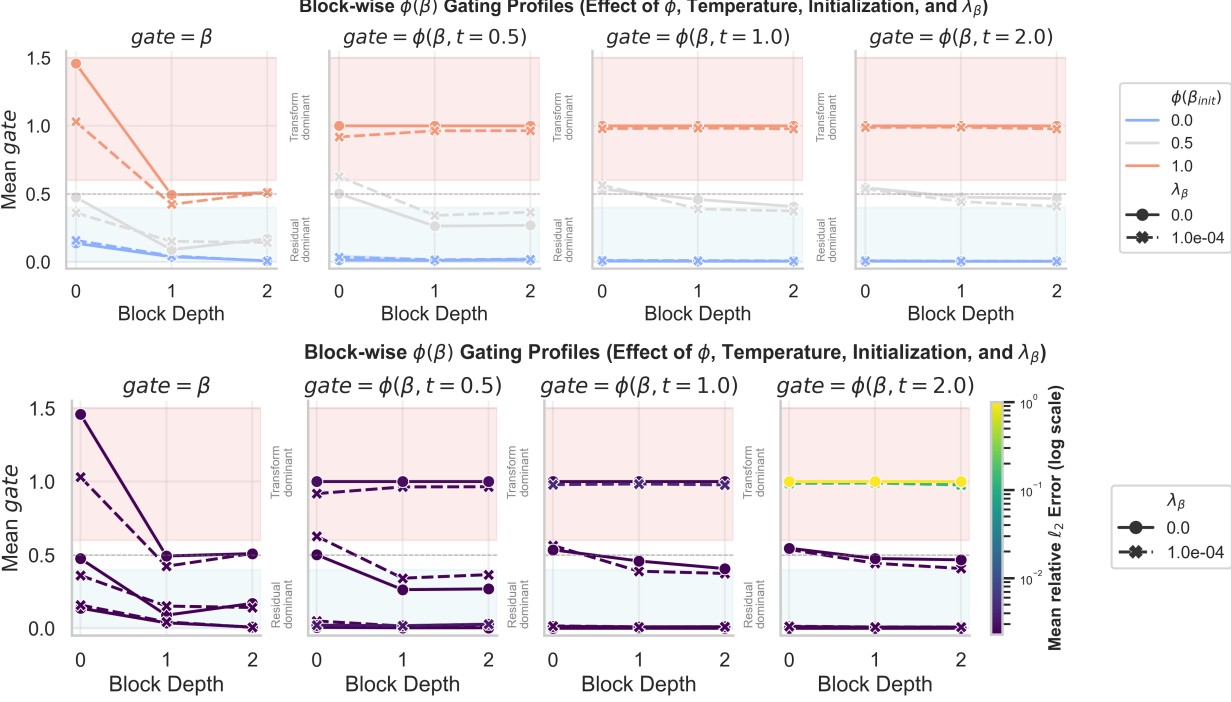

Figure 26: *Allen-Cahn equation:* Depth-wise $\beta$ gating parameters for varying initialization, temperature, and regularization. The $\beta$-gates modulate residual information flow across depth. Smaller gate values bias toward identity mapping (residual-dominant), and larger gate values increase transform-dominant behavior.

contributions, but also dynamically regulate the effective network depth, enabling HyResPINNs to self-adjust their representational complexity according to the PDE structure and training regime.

## B   Allen-Cahn with Periodic Boundary Conditions

### B.1   Reference Solution Generation

Following the approach in (Wang et al., 2024), we employ standard spectral methods to solve the Allen–Cahn equation under periodic boundary conditions. Starting from the initial state $u_0(x) = x^2 \cos(\pi x)$, we integrate the equation up to a final time of $T = 1$. For generating synthetic validation data, we use the Chebfun (Driscoll et al., 2014) package to perform a Fourier spectral discretization with 512 modes. Time integration is carried out using the fourth-order exponential time-differencing Runge–Kutta method (ET-DRK4) (Cox & Matthews, 2002) with a time step of $10^{-5}$. Solutions are sampled every $\Delta t = 0.005$, resulting in a high-resolution validation dataset of size $200 \times 512$.

## C   Darcy Flow with Smooth Coefficients

### C.1   Learned Gating Parameter Evolution

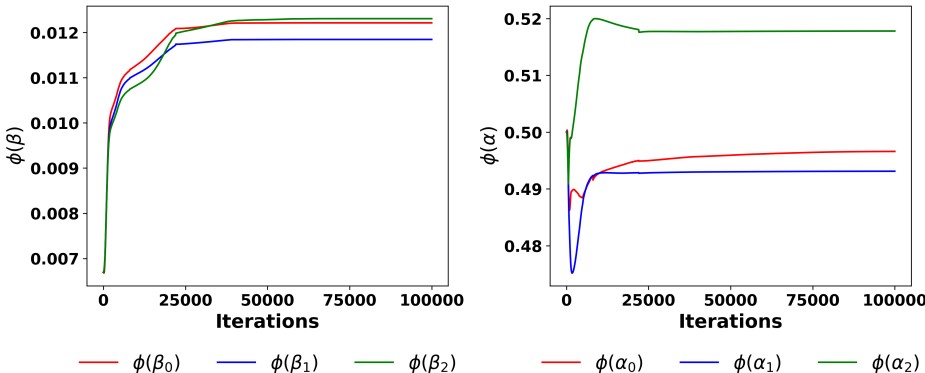

Figure 27: *2D Smooth Darcy-Flow equation (Neumann BCs):* Evolution of the learned gating parameters during training. (Left) Evolution of residual gating parameters $\phi(\beta_i)$, which modulate the strength of residual information flow across blocks. (Right) Evolution of hybrid gating parameters $\phi(\alpha_i)$, which balance the DNN and RBF components within each block.

## D   Kuramoto-Sivashinsky Equation

### D.1   Reference Solution Generation

We employ standard spectral methods to solve the Kuramoto-Sivashinsky equation under periodic boundary conditions. For generating synthetic validation data, we use the Chebfun (Driscoll et al., 2014) package to perform a Fourier spectral discretization with 512 modes. Time integration is carried out using the fourth-order exponential time-differencing Runge–Kutta method (ETDRK4) (Cox & Matthews, 2002) with a time step of $10^{-5}$. Solutions are sampled every $\Delta t = 0.004$, resulting in a high-resolution validation dataset of size $250 \times 512$.

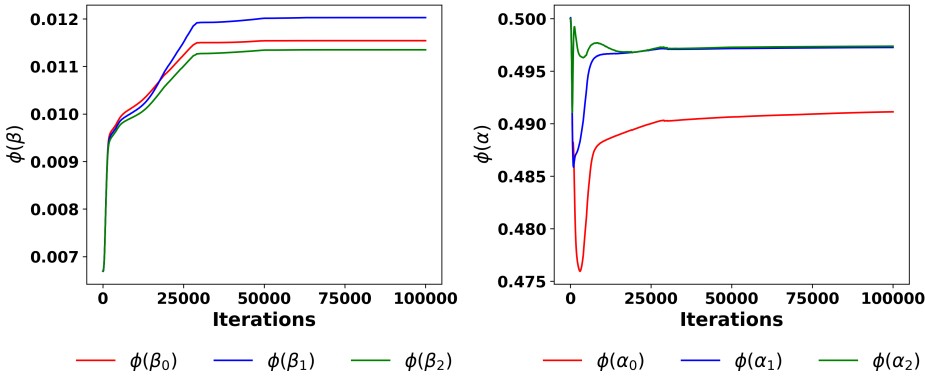

Figure 28: *3D Smooth Darcy-Flow equation (Dirichlet BCs):* Evolution of the learned gating parameters during training. (Left) Evolution of residual gating parameters $\phi(\beta_i)$, which modulate the strength of residual information flow across blocks. (Right) Evolution of hybrid gating parameters $\phi(\alpha_i)$, which balance the DNN and RBF components within each block.

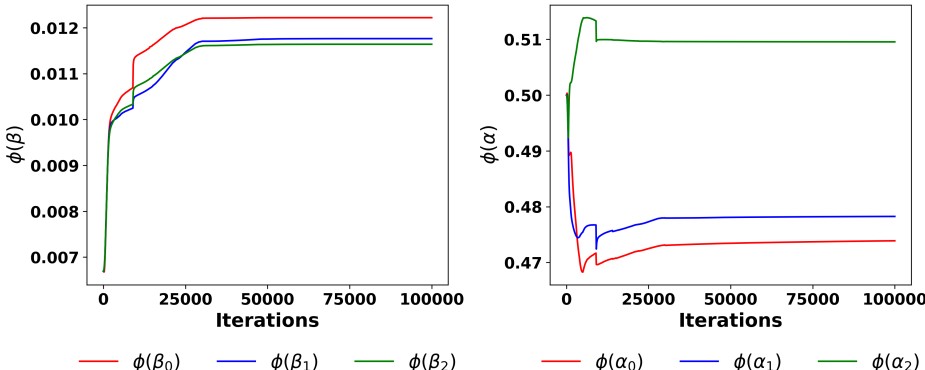

Figure 29: *3D Smooth Darcy-Flow equation (Neumann BCs):* Evolution of the learned gating parameters during training. (Left) Evolution of residual gating parameters $\phi(\beta_i)$, which modulate the strength of residual information flow across blocks. (Right) Evolution of hybrid gating parameters $\phi(\alpha_i)$, which balance the DNN and RBF components within each block.

# E    Korteweg-De Vries Equation

## E.1    Reference Solution Generation

Similar to (Wang et al., 2024), we generate synthetic validation data for the Korteweg–De Vries equation, we use standard spectral methods under periodic boundary conditions. The simulation begins with the initial condition set to $u_0(x) = \cos(\pi x)$ and is integrated forward in time up to $T = 1$. The Chebfun package (Driscoll et al., 2014) is utilized for a Fourier spectral discretization employing 512 modes. Time integration is performed with the fourth-order exponential time-differencing Runge–Kutta (ETDRK4) method (Cox & Matthews, 2002), using a step size of $10^{-5}$. The solution is sampled every $\Delta t = 0.005$, resulting in a validation dataset of size $200 \times 512$ covering the temporal and spatial domains.

# F    Grey-Scott Equation

## F.1    Reference Solution Generation

For data generation, we solve the Grey-Scott equation using standard spectral techniques under periodic boundary conditions following the details provided in (Wang et al., 2024). The system is initialized with the prescribed initial condition and integrated up to a final time of $T = 2$. Synthetic validation data are produced with the Chebfun package (Driscoll et al., 2014), employing a 2D Fourier spectral discretization with $200 \times 200$ modes. Temporal integration is performed using the fourth-order exponential time-differencing Runge–Kutta (ETDRK4) method (Cox & Matthews, 2002) with a time step of $10^{-3}$. Solutions are saved every $\Delta t = 0.02$, resulting in a validation dataset of dimensions $100 \times 200 \times 200$ spanning both the temporal and spatial domains.

