# OpenReview forum: "HyResPINNs: A Hybrid Residual Physics-Informed Neural Network Architecture Designed to Balance Expressiveness and Trainability"
_TMLR — Accepted by TMLR_

### Review · Reviewer_CAVd · 2025-10-01

**Summary Of Contributions:**

The paper introduces HyResPINNs, an architecture that combines PINNs with radial basis functions (RBFs) to address the tendency of PINNs to produce oversmoothed solutions. The method demonstrates promising performance across diverse benchmark problems, with rigorous comparisons conducted under a fixed parameter budget. An ablation study further shows that the RBF and PINN components complement each other positively.

A key strength of the work is the thorough evaluation, including benchmarks and ablations, which provide evidence for the benefits of the proposed hybrid design. A weakness is the increased architectural complexity: the additional RBF network introduces more hyperparameters and may complicate training, though the reported training curves appear robust. A further limitation is that the approach may be less beneficial for PDEs with inherently smooth solutions.

**Additional Comments:**

There is a typo in Figure 3(a): It should be "left" instead of "lef".

**Audience:**

Yes

**Audience Explanation:**

Yes, the findings of this paper would be of interest to the TMLR audience. The work represents a meaningful contribution at the intersection of machine learning and computational science, particularly for PDE problems with sharp gradients where standard PINNs often struggle. More generally, the proposed approach provides a helpful inductive bias in the parametrization of neural networks for addressing such problems.

**Broader Impact Concerns:**

The paper does not include a broader impact statement. I do not see any pressing ethical or societal concerns arising from this work. The contribution is primarily methodological within the domain of scientific machine learning, and I see little potential for negative impact.

**Claims And Evidence:**

Yes

**Claims Explanation:**

The claims in the submission are generally well supported by accurate and convincing evidence. The authors present results on seven subproblems, showing that HyResPINNs outperform meaningful benchmarks and providing substantial experimental evidence for the benefits of the architecture. The experiments are run over multiple random seeds, which strengthens the statistical significance of the findings.

While it is possible that there are some cases where the architecture does not outperform, particularly for problems with inherently smooth solution functions, the presented evidence is strong and demonstrates clear improvements for problems with sharp gradients in the solution function.

**Requested Changes:**

In the Related Works section, the differences relative to the most similar methods are left largely implicit. The paper would benefit from a concrete contrasts that states how HyResPINN differs from each closest prior work (e.g., brief one-sentence “X does …; HyResPINN differs by …” statements) either at the end of Related Work or in the “Positioning” paragraph in the Introduction. This will also place the chosen baselines in clearer context and help readers interpret the experimental comparisons and the scope of the claimed contributions.

To strengthen the experimental support, it would be helpful to report standard deviations across seeds in Table 2 and to mark the second-best results (e.g., with underlining).

It would also be valuable to visualize how the gating mechanism behaves at different network depths, as this could inspire architectural variants that fix these values from the start and would further demonstrate that both components are actively used. Hereby, I would encourage the authors to create a plot with the network depth as the x-axis and the value of the learned gating weight on the y-axis, over multiple seeds, to see where the network uses which part of the Hybrid Residual Block.

Finally, adding a clear limitations section would improve the paper: for example, discussing when this architecture might struggle or cases where a regular PINN might be expected to outperform.

---

> ### Author Response · Authors · 2025-10-24
> **Response to Reviewer CAVd**
>
> We thank the reviewer for their constructive feedback. We appreciate the reviewer’s recognition of the paper’s contributions and have carefully addressed all requested changes. We have added a brief “Broader Impact” section at the end of the paper and have corrected the minor typos. Below, we summarize our responses and corresponding revisions.
>
> > Related Works.
>
> We thank the reviewer for this suggestion. We have now explicitly contrasted HyResPINNs with each of the selected baselines—Standard PINN, ResPINN, ExpertPINN, PirateNet, and StackedPINN—in both Section 2: Positioning of HyResPINNs and Section 5: Baseline Methods. We describe each method alongside a clear statement of how HyResPINNs differ, as suggested, highlighting the specific architectural mechanism added at each step and the scientific question it addresses. This revision clarifies the experimental scope and the rationale for the chosen baselines.
>
> > To strengthen the experimental support, it would be helpful to report standard deviations across seeds in Table 2 and to mark the second-best results (e.g., with underlining).
>
> We thank the reviewer and agree that this additional result clarification will aid in the experimental result reporting. We have marked the second-best results in Table 2. To avoid overcrowding Table 2 with excessive information, we report the corresponding standard deviations in the problem-specific tables within each experiment section. Specifically, Tables 3–9 have been updated to include both the error values and their standard deviations, along with the additional evaluation metrics introduced following Reviewer vFVz’s recommendation.
>
> > It would also be valuable to visualize how the gating mechanism behaves at different network depths, as this could inspire architectural variants that fix these values from the start and would further demonstrate that both components are actively used...
>
> We thank the reviewer for this suggestion. We have updated the manuscript to include an ablation study in Appendix A.3 on the gating parameters using different initializations. Also, Figures 4, 9, 13, 15, 18, 20, 22, 27, 28,  and 29 show the learned α and β gating parameters across depth and training iteration. Overall, we observe that both components (RBFNN and DNN) are actively used throughout training. During the early stages of training, alpha exhibits a broad range of values, but typically converges toward values that slightly favor the RBFNN within each block. Conversely, the beta gating values initialized near 0 gradually increase during training, reflecting a smooth shift from predominantly identity flow toward stronger nonlinear transformation as deeper features are learned.
>
> > Limitations.
>
> We thank the reviewer for this suggestion. We have included a paragraph titled “Limitations and Future Work” within Section 6 (Conclusion). This section discusses scenarios where HyResPINNs may be less beneficial than other architectures, the increased hyperparameter sensitivity and computational overhead due to the additional RBF network, and future research directions for simplifying the architecture and reducing hyperparameter complexity. These additions ensure that the paper presents a balanced and transparent discussion of both strengths and limitations.

---

### Review · Reviewer_yiAS · 2025-10-02

**Summary Of Contributions:**

This paper introduces hybrid residual physics-informed neural networks, HyResPINNs, a new architecture to solve PDEs in a mesh-free data driven manner. The authors motivate this new architecture by decomposing expressivity into three parameters: function space richness, adaptivity, and efficiency per DoF, over which they situate their method and baselines to analyze contributions and limitations. By combining the efficiency and adaptivity of classical methods through the use of local RBF neural networks and skip connections with the richness of neural networks through deep neural networks within each block and at the end of the architecture, HyResPINNs aim at addressing the three axes at once. The authors motivate each architectural and training choice with a wide range of literature references, and demonstrate the performance of various PDE experiments.

**Additional Comments:**

### Recommendation

Once my remarks have been addressed, I will be leaning toward accepting this paper, considering its sound results and its potential for the PINN community.

**Audience:**

Yes

**Audience Explanation:**

This method will definitely be of interest to the community. PINNs are an active field of research with wide applications, and these results against recent state-of-the-art methods will definitely contribute to the field. Future directions, including experiments on higher-dimensional problems, deeper comparisons over different classes of methods, definitely deserve attention.

**Broader Impact Concerns:**

None.

**Claims And Evidence:**

Yes

**Claims Explanation:**

The paper is well written and is pleasant to read. There are very few typos, it is well-structured and well-divided. I enjoyed the explanation of the three axes of comparison for efficiency, and how the overall discussion follows them.

The method itself is well-supported by the literature. Each design choice is properly motivated with references. They are logically connected to each goal of the paper and the evidence provided by the experiments, notably the ablation studies, further supports the overall architecture.

The experiments are diverse, encompass 1D to 3D for the spatial domain. I think the choice of the baselines is appropriate, but could be slightly enhanced to further support the claims of the paper against classical and general ML methods.

### Questions
- When describing your method, you support the gating mechanism for its ability to selectively adapt the required depth of the network depending on the problem, in contrary to other overparameterized neural networks. In practice, your formulation of the model uses a gated external skip connection between each subsequent blocks. Did you observe, in your experiments, models which properly used only a subset of the block, e.g.,  some phi(betas) being null for the last m blocks? Did you experiment with skip connections without gating? I find the identity initialization interesting, maybe I missed some details but were some other initializations experimented with? It would be interesting to compare between gated and ungated skip connections as well as different initializations to support their impact beyond gradient stability.
- Although I understand the focus on PINNs, the claims of the paper address more general comparisons with classical methods and especially different ML classes of architectures. Have you had the opportunity to train other classes of models, such as FNOs, GNOT, GeoFNO, or GINO? It would be interesting, even just for indicative purposes. I understand this can be too much work, so I am not really asking for new experiments but rather for possibly already conducted ones, expectations, or reasons not to have done them.
- Although some of the PINN baselines are motivated because they were described in the related work section, I think a deeper baselines paragraph which further explains the reasons for each of them, especially regarding the three axes of expressivity being tackled, could help reading and interpreting Section 5.
- Table 4 & Fig 8: HyResPINN has similar mean relative L2 error over training compared to Expert & Pirate, yet is respectively 37 and 19% slower. I am not sure these times should be considered as really comparable in the text, in contrary to the errors in this case.

**Requested Changes:**

First, please address my questions in the first section. The next points are minor writing-related remarks.

- I'm not really considering it strictly required, but why do only some subsections in Section 5 have a "summary"? This is only a question regarding writing style, I was expecting sort of the same structure for each experiment. This is minor.

### Typos

These are only some minor typos I found when reading.

- The citation of "The p and h-p Versions of the Finite Element Method, Basic Principles and Properties" has a bold v in Babuška, which appears in the text.
- Penultimate paragraph of page 4 (starting with "For instance"), end of line 3: double "potentially".
- Section 3.2, end of first line, missing "in" between "resulting" and "training".
- Typo in Jax version? 0.7.2 is the latest as far as I know and you wrote 1.11.

---

> ### Author Response · Authors · 2025-10-24
> **Response to Reviewer yiAS (1/2)**
>
> We thank the reviewer for their constructive feedback. We have carefully addressed all requested changes. The requested clarifications on gating, baselines, and efficiency have led to substantial improvements in the paper’s rigor and clarity. The additional experiments, expanded discussions, and revised results presentation directly address all points raised. Further, we have included Limitations and Future work paragraphs in the Conclusion to discuss the potential future directions mentioned, including experiments on higher-dimensional problems and further comparisons with other methods. Below, we summarize our responses and corresponding revisions.
>
> > When describing your method, you support the gating mechanism for its ability to selectively adapt the required depth of the network depending on the problem, in contrary to other overparameterized neural networks...
>
> We thank the reviewer for raising these questions regarding the trainable gating parameters. We have updated the manuscript to include an ablation study on the gating parameters using different initializations in Appendix A.3. Also, Figures 4, 9, 13, 15, 18, 20, 22, 27, 28,  and 29 show the learned α-gate β-gate values across depth and training iteration. These results illustrate the learned gating parameter values across depth, initialization, temperature, and regularization settings. Our findings show that:
> - The identity initialization of the β gates effectively regulates the residual pathway. However, we did not find that deeper blocks learn β values near zero, indicating that all residual paths remain active to some extent rather than being entirely skipped.
> - The α-gates, which balance the RBFNN and DNN contributions, adapt the representation ratio to slightly prefer the RBFNN outputs and converge to a fixed ratio across each block in most cases.
> - While we did not test disabling the gating mechanism completely, we tested variants without the sigmoid transformation of the gating parameters. This removes the bounded nonlinear gating and allows unrestricted scaling of the residual and identity contributions (and similarly for the RBFNN and DNN components). Although this configuration does not correspond to strictly “ungated” skip connections, it approximates the behaviour of an unmodulated residual path, specifically, when β remains near identity values, the residual path behaves similarly to an ungated connection. We observed similar error results in these cases compared to the sigmoidal gates, however, the sigmoidal formulation offers an interpretability advantage, as the learned gate values can be directly interpreted as the relative contribution of each pathway, bounded between 0 and 1.
> - We further tested alternative initializations for both alpha and beta and found that the identity initialization for beta consistently provides the best error results, largely independent of the alpha initialization.
>
> > Although I understand the focus on PINNs, the claims of the paper address more general comparisons with classical methods and especially different ML classes of architectures...
>
> We appreciate this suggestion and agree that operator-learning architectures such as FNOs, GNOT, GeoFNO, GINO, and others, such as DeepONets, generalize the single-instance PDE solvers focused on in this work. Given the length of the current work, we decided to focus specifically on solving single-instance PDE solutions through the PINN framework of PDE residual minimization, rather than operator-learning models that learn mappings between function spaces. Nonetheless, we included a brief discussion on operator learning models in Section 2: Related Works and added an explicit Future Works section in the Conclusion, where we mention extensions to the HyResPINN scope.
>
> > Regarding the baseline methods motivation.
>
> We thank the reviewer for this suggestion. In alignment with our updates in response to Reviewer CAVd’s comment requesting more explicit contrasts with related methods, we have revised Section 2: Positioning and Section 5: Baseline Methods to more explicitly motivate and contextualize each of the chosen baselines. Specifically, we now:
> - Added explicit contrasts at the end of the Related Work section, clearly distinguishing HyResPINNs from the most similar prior architectures (Standard PINN, ResPINN, ExpertPINN, PirateNet, StackedPINN).
>  -Expanded the Baseline Methods subsection to explain the scientific question each baseline addresses and how each incrementally explores one or more axes of expressivity, adaptivity, and efficiency per degree of freedom.
>
> Together, these revisions ensure that the baseline choices are now explicitly tied to the paper’s conceptual framework.

---

> ### Author Response · Authors · 2025-10-24
> **Response to Reviewer yiAS (2/2)**
>
> > Tables 4 and Figure 8.
>
> We agree with the reviewer’s observation and have revised the discussion accompanying Table 4 and Figure 8 (now Figure 10) to clarify that, although HyResPINNs can sometimes exhibit higher computational cost than the tested baselines, they achieve superior overall performance when accounting for the number of trainable parameters and the error obtained. Specifically, Section 6 now includes explicit quantitative metrics for efficiency per degree of freedom ($\eta_{DoF}$, $\eta_{comp}$, and $\eta_{params}$), providing more rigorous performance measures that jointly account for model complexity, computational time, and predictive accuracy. These metrics show that HyResPINNs achieve higher efficiency relative to their computational cost and representational capacity.
>
> > Typos.
>
> We thank the reviewer for catching these inconsistencies. We have standardized the structure of Section 6 by standardizing each sub-paragraph within each subsection. We have also corrected all noted typographical issues.

---

### Review · Reviewer_vFVz · 2025-10-04

**Summary Of Contributions:**

This paper aims to enhance the function space richness, adaptivity and efficiency per degree of freedom of physics-informed neural networks (PINNs) for solving PDEs. Specifically, it introduces a hybrid residual PINN architecture that combines deep neural networks with radial basis function neural networks, using a learnable gating parameter to adaptively balance the contributions of these two modules. Besides, a second learnable gating parameter is used to modulate the residual connection, enabling dynamic adaptation of network depth during training.

The proposed architecture is evaluated on a variety of PDEs with different input dimensions and boundary conditions, including cases with sharp solution features. It demonstrates higher accuracy than baseline methods, indicating the effectiveness of the hybrid design.

**Audience:**

Yes

**Audience Explanation:**

This work addresses the challenge of solving PDEs with deep neural networks and aims to improve the accuracy and expressiveness of PINN-based approaches, which should be a topic of broad interest.

**Broader Impact Concerns:**

There are no broader impact concerns.

**Claims And Evidence:**

No

**Claims Explanation:**

- The definition of efficiency per degree of freedom (DoF) is unclear, and no quantitative metric is provided to evaluate this property. In the paper, efficiency per DoF is illustrated through plots of mean relative L2 error with respect to training iterations or training set size. While these results indicate that the proposed HyResPINN architecture converges faster to a lower error compared to other baseline methods, this does not constitute a quantitative measure of efficiency per DoF. Moreover, as shown in Tables 3 and 4, HyResPINN does not achieve the lowest computational cost. Therefore, the claim of 'superior efficiency per DoF' is not fully supported. Instead, the results suggest that HyResPINN provides a reasonable trade-off between computational cost and accuracy.

- In Section 5.3, “Efficiency per Degree of Freedom”, the authors claim that 'HyResPINN consistently achieves lower solution and flux errors than competing models, especially in the low-data regime.' However, in the first row of Figure 10, HyResPINN does not achieve lower solution and flux errors than the competing models; in fact, ResPINN attains a lower solution error than HyResPINN at the lowest training set size for Dirichlet boundaries. I suggest the authors revise this statement to ensure greater accuracy and rigor in their scientific expressions.

**Requested Changes:**

- It would be good if the authors could further analyze the values of the two learnable gating parameters (one controlling the weighting between the DNN and RBFNN, and the other controlling the weighting of the residual module) for different PDEs, to investigate if they have some physical interpretations or insights.

- In table 2, for DarcyFlow with 3D Dirichlet BCs boundary conditions, ResPINN achieves the same L2 test error as HyResPINN and should also be highlighted in bold. Similarly, ExpertPINN should be highlighted for DarcyFlow 3D with Neumann boundary conditions.

---

> ### Author Response · Authors · 2025-10-24
> **Response to Reviewer vFVz**
>
> We thank the reviewer for their constructive feedback. The reviewer’s comments have led to several key improvements: a formalized efficiency framework, a more accurate and balanced presentation of results, and a more in-depth analysis of the learnable gating mechanisms. Below, we summarize our responses to each of the reviewer’s points and describe the corresponding revisions.
>
> > The definition of efficiency per degree of freedom (DoF) is unclear, and no quantitative metric is provided to evaluate this property.
>
> We thank the reviewer for these critical comments and agree that our previous discussion of efficiency per DoF was primarily qualitative. In response, we have revised Section 6 to formally define this concept and introduce explicit quantitative metrics ($\eta_{\text{params}}$, $\eta_{\text{comp}}$, and $\eta_{\text{DoF}}$) that jointly account for model accuracy, capacity, and computational cost. These new metrics provide empirical measures for assessing the trade-offs between accuracy, model capacity, and runtime.
>
> We emphasize that these metrics are not derived from a theoretical efficiency formulation. Instead, they serve as practical indicators for analyzing trade-offs between accuracy, expressivity, and computational cost. Using these definitions, we have reanalyzed our results and find that HyResPINNs achieve the highest overall efficiency ($\eta_{\text{DoF}}$) among tested architectures, in most cases, confirming that the hybrid design achieves a favorable balance between expressivity and efficiency. Additionally, we now note that while HyResPINNs incurs a modest increase in computational cost in the reported tables, it provides improved accuracy in most cases, which is captured quantitatively through these empirical efficiency measures.
>
> > In Section 5.3, “Efficiency per Degree of Freedom”, the authors claim that 'HyResPINN consistently achieves lower solution and flux errors than competing models, especially in the low-data regime.'
>
> We thank the reviewer for noting this oversight. We have revised the relevant sentence in Section 6 to be more accurate. We also added a discussion to explicitly acknowledge the few cases (e.g., ResPINN under Dirichlet boundaries) where HyResPINN achieves comparable or worse rather than superior accuracy. These changes ensure that all claims are fully consistent with the reported data.
>
> > It would be good if the authors could further analyze the values of the two learnable gating parameters.
>
> We thank the reviewer for this suggestion. In response, we have added new analyses examining the learned α- and β-gating profiles across depth, initialization, temperature, and regularization settings (see our related comments to Reviewers CAVd and yiAS). These results provide a comprehensive view of how the gating parameters evolve during training for different PDEs. Further, we did not find any clear patterns across PDE types that could lead to physical insights, though these types of connections would be an interesting avenue to explore in future works.
>
> > Regarding Table 2.
>
> We thank the reviewer for catching this detail. We have corrected Table 2 accordingly, ensuring that all cases with tied or near-identical performance are consistently highlighted in bold. We have also reviewed all tables in the paper to verify formatting consistency for best and second-best results.

---

### Author Response · Authors · 2025-10-24
**Summary response to all reviewers**

We thank all reviewers for their helpful and constructive feedback. The comments have improved the depth, clarity, and overall quality of our paper. While we recognize that a reviewer discussion period was available, we found the reviews to be clear, actionable, and consistent in scope, requiring no additional clarification from our side.

In the revised manuscript, we indicate responses and corresponding edits using color for clarity. Specifically, comments and edits addressing Reviewer CAVd are shown in blue, Reviewer yiAS in orange, Reviewer vFVz in red, and any additional revisions are shown in purple. When multiple reviewers raised similar points, we used the color of the first reviewer to identify the issue.

We will address each reviewer's comments in detail in individual responses. There were several overlapping points raised by the reviewers, which we summarize below.

## 1. (Analysis of the gating parameters)
All reviewers encouraged a deeper investigation into the gating mechanisms of the HyResPINN architecture that modulate both the (i) residual connection strength across the hybrid residual blocks and (ii) the relative contribution of the DNN to RBFNN components within each block.

- Specifically, Reviewer CAVd requested a visualization of how the gating parameters evolve across network depth and training.
- Reviewer yiAs suggested, including ablations comparing gated and ungated skip connections as well as different initialization schemes.
- Reviewer vFVz requested further analysis of the learned α and β parameters across different PDEs to explore whether they exhibit interpretable patterns related to underlying physical properties.

In response, we have added new experiments and visualizations illustrating the learned gating values across each residual block. Specifically, Figures 4, 9, 13, 15, 18, 20, 22, 27, 28,  and 29 show the evolution of the learned gating parameters during training, and Appendix A.3 details the mean gating behavior across residual blocks for different initialization and regularization strategies. Specifically, this section details ablation experiments comparing gated and ungated variants, as well as regularized and unregularized gating parameters. These results clarify how the gating parameters evolve during training, how they differ across different PDEs, and how different initializations and regularization strengths influence the learned gates and final results.

## 2. (Clarity and completeness of experimental reporting)
Reviewers requested clearer reporting of the experimental results, including standard deviations and highlighting of second-best results. We have revised the tables accordingly. In particular, Tables 3-9 now include standard deviations across multiple random seeds, and Table 2 now marks the second-best results for easier comparison.

## 3. (Efficiency per degree of freedom clarification and formalization)
Reviewer vFVz noted that our prior discussion of “efficiency per degree of freedom” was primarily qualitative. In response, we have now formalized this concept by introducing explicit metrics, including:
- Parameter efficiency ($\eta_{\text{params}}$): the inverse of the product of the relative $\ell_2$ error and the number of trainable parameters, quantifying the accuracy achieved per degree of freedom.
- Computational efficiency ($\eta_{\text{comp}}$): the inverse of the product of the relative $\ell_2$ error and the average training time (represented as the number of training iterations completed per second), representing the accuracy achieved per unit computational cost.
- Efficiency per DoF ($\eta_{\text{DoF}}$): a combined metric that integrates both the number of trainable parameters and computational cost, enabling comparison across model architectures.

These metrics provide a more rigorous basis for analyzing the trade-offs between accuracy and computational cost and confirm that HyResPINNs achieve a favorable balance compared to standard PINNs and other baselines.

## 4. (Clarifications regarding related methods and the positioning of HyResPINNs)
Reviewers CAVd and yiAS requested clearer differentiation between the proposed HyResPINNs and related architectures. In response, we have revised the “Positioning of HyResPINNs” paragraph in Section 2 (Related Work) and the “Baseline Methods” paragraph in Section 5 (Experimental Results and Discussion) to include a clearer discussion on how the proposed methods differ from related approaches.

## Summary.
We believe these revisions address all reviewer concerns thoroughly and strengthen both the technical rigor and explanatory aspects of the paper. The added experiments and visualizations enhance the interpretability of the proposed method and provide a deeper understanding of how hybrid gating contributes to the performance improvements. We thank the reviewers again for their insightful feedback and constructive guidance.

---

### Decision · Action_Editor_Kqsq · 2025-11-03

**Recommendation:** Accept as is

**Additional Comments:**

Please make sure to cleanly incorporate all the changes arising from the reviewer author discussion in the final version.

**Audience:**

Yes

**Audience Explanation:**

PINNs are of interest to a sizable community within the core machine learning audience and the presented architecture is certainly of interest to that community.

**Claims And Evidence:**

Yes

**Claims Explanation:**

The presented PINN approach is driven by well-motivated design choice motivation and is extensively evaluated empirically. The authors extensively responded to reviewers initial feedback and clarified initial questions and addressed concerns well.